 

# Universality of clonal dynamics poses fundamental limits to identify stem cell self-renewal strategies

**Cristina Parigini[1,2], Philip Greulich[1,2]***

[1]School of Mathematical Science, University of Southampton, Southampton, United Kingdom; [2]Institute for Life Sciences, University of Southampton, Southampton, United Kingdom

**Abstract** How adult stem cells maintain self-renewing tissues is commonly assessed by analysing clonal data from *in vivo* cell lineage-tracing assays. To identify strategies of stem cell self-renewal requires that different models of stem cell fate choice predict sufficiently different clonal statistics. Here, we show that models of cell fate choice can, in homeostatic tissues, be categorized by exactly two 'universality classes', whereby models of the same class predict, under asymptotic conditions, the same clonal statistics. Those classes relate to generalizations of the canonical asymmetric vs. symmetric stem cell self-renewal strategies and are distinguished by a conservation law. This poses both challenges and opportunities to identify stem cell self-renewal strategies: while under asymptotic conditions, self-renewal models of the same universality class cannot be distinguished by clonal data only, models of different classes can be distinguished by simple means.

***For correspondence:**
P.S.Greulich@soton.ac.uk

**Competing interests:** The authors declare that no competing interests exist.

## Introduction

Adult stem cells are the key players for maintaining and renewing biological tissue, due to their ability to persistently produce tissue cells through cell division and differentiation (***National Institute of Health, 2009***). For maintaining tissues in a homeostatic state, it is crucial that stem cells adopt suitable *self-renewal strategies*, a pattern of stem cell fate choices that balances proliferation and differentiation; otherwise, imbalanced proliferation may lead to hyperplasia and cancer. Therefore, the understanding and identification of stem cell self-renewal strategies has been a major goal of stem cell biology ever since the discovery of adult stem cells.

Classically, two stem cell self-renewal strategies have been proposed (***Potten and Loeffler, 1990***; ***Simons and Clevers, 2011a***): following the *Invariant Asymmetric division* (IA) strategy, stem cells undertake only asymmetric divisions, whose outcome is one differentiating cell and one stem cell as daughter cells. The other proposed strategy, *Population Asymmetry* (PA) (***Potten and Loeffler, 1990***; ***Simons and Clevers, 2011a***; ***Watt and Hogan, 2000***; ***Klein and Simons, 2011***), features additionally symmetric divisions, which produce either two stem cells or two differentiating cells as daughters, yet in balanced proportions. Both patterns of cell fate choice leave the number of cells on average unchanged and thus can maintain homeostasis. Assessing stem cell self-renewal strategies experimentally is difficult in vivo, since direct observation of cell divisions is rarely possible. Yet, through genetic cell lineage-tracing assays, the statistics of clones – the progeny of individual cells – can be obtained, and via mathematical modeling assessing cell fate dynamics became possible. With such an approach several studies suggested that population asymmetry prevails in many mouse tissues (e.g. ***Clayton et al., 2007***; ***Lopez-Garcia et al., 2010***; ***Simons and Clevers, 2011b***; ***Doupé et al., 2012***; ***Klein et al., 2010***).

However, the interpretation of those studies has been challenged by a suggested alternative self-renewal strategy, called *Dynamic Heterogeneity* (DH), featuring some degree of cell fate plasticity (*Greulich and Simons, 2016*). In this model, all stem cell divisions are asymmetric, yet it is in agreement with the experimental clonal data that had previously been shown to agree also with the population asymmetry strategy. Thus, those two strategies are not distinguishable in view of the clonal data.

This raises the question to what extent different stem cell self-renewal strategies can be distinguished at all via clonal data (*Klein and Simons, 2011*; *Greulich, 2019*). Here, we address this question by studying models for stem cell fate choice, which define the self-renewal strategies, in their most generic form. We show that many cell fate models predict, under asymptotic conditions, the same clonal statistics and thus cannot be distinguished via clonal data from cell lineage-tracing experiments. In particular, we find that there exist two particular classes of stem cell self-renewal strategies: one class of models which all generate an Exponential distribution of clone sizes (the number of cells in a clone) after sufficiently large time, and one which generates a Normal distribution under sufficiently fast stem cell proliferation. Crucially, these two classes are not differentiated via the classical definitions of symmetric and asymmetric stem cell divisions, but by whether or not a subset of cells is conserved. These classes thus bear resemblance to 'universality classes' known from statistical physics, as suggested in *Klein and Simons, 2011*. This leads us to a more generic, and in this context more useful, definition of the terms 'symmetric' and 'asymmetric' divisions. Notably, however, we find that the conditions for the emergence of universality are not always fulfilled in real tissues, which provides chances, but also further challenges, for the identification of stem cell fate choices in homeostatic tissues.

## Strategies for stem cell self-renewal

The two classical stem cell self-renewal strategies, Invariant Asymmetry (IA) and Population Asymmetry (PA) (*Potten and Loeffler, 1990*; *Simons and Clevers, 2011a*; *Watt and Hogan, 2000*; *Klein and Simons, 2011*), are commonly described in terms of two cell types: stem cells (*S*) which can self-renew (i.e. divide without reducing their potential to divide in the future); and differentiating cells (*D*). Both strategies can be expressed in terms of a single parametrized stochastic model, a multi-type branching process (*Haccou et al., 2005*), defined by the outcomes of cell divisions (the *cell fate choices*),

$$S \xrightarrow{\lambda} \begin{cases} S+S & \text{with probability } r \\ S+D & \text{with probability } 1\text{-}2r, \\ D+D & \text{with probability } r \end{cases} \qquad (1)$$

where cells of type *S* divide with rate $\lambda$. Here, a daughter cell configuration $S+S$ corresponds to *symmetric self-renewal division* and $D+D$ to *symmetric differentiation*, while daughter cells of different type, $S+D$, marks an *asymmetric division*. In the basic model version, a cell of type *D* is eventually lost with rate $\gamma$, $D \xrightarrow{\gamma} \emptyset$ (corresponding to death, shedding, or emigration of *D*-cells), while other versions may include the possibility of limited proliferation as committed progenitor cells. The two self-renewal strategies, IA and PA, are distinguished by the value of the *symmetric division fraction r*: the PA model corresponds to any $0 < r \leq \frac{1}{2}$; the IA model is defined by $r = 0$, that is, only asymmetric divisions occur.

To maintain homeostasis, the number of cells must stay, on average, constant. Thus cells following the PA strategy must regulate the probabilities of symmetric self-renewal and differentiation to be exactly equal, whereas for the IA model this is trivially assured. However, only for the IA model is the number of stem cells *strictly conserved*, that is, no gain or loss of stem cells is possible.

A way to assess self-renewal strategies experimentally is via genetic cell-lineage tracing (*Kretzschmar and Watt, 2012*; *Blanpain and Simons, 2013*): By marking single cells with an inheritable genetic marker (through a Cre-Lox system [*Soriano, 1999*; *Sauer, 1998*]) each cell's progeny, called a *clone*, which retain that marker, can be traced. The number of cells per clone, that is the *clone size*, is measured and the statistical frequency distribution of clone sizes (*clone size distribution*) determined. To test the cell fate choice models on that data, one evaluates the models with a single cell as initial condition and samples the outcome in terms of the final cell numbers – the size

of a virtual clone. In the basic version of the model (i.e. when $D \xrightarrow{\gamma} \emptyset$), the IA and PA models predict, respectively, a Poisson and an Exponential clone size distribution for large times (**Klein and Simons, 2011**; **Antal and Krapivsky, 2010**) (see also the Appendix, 'Invariant Asymmetry and Population Asymmetry models'). Thus, they are fundamentally different and can easily be distinguished when compared with clonal data. By a series of lineage-tracing experiments it was confirmed that Exponential clone size distributions prevail for most mouse tissues, which thus exclude the IA model and support the PA strategy (**Clayton et al., 2007**; **Lopez-Garcia et al., 2010**; **Simons and Clevers, 2011b**; **Doupé et al., 2012**; **Klein et al., 2010**).

While this seemed to settle the case in favour of the PA strategy, at least for most adult mouse tissues, this was challenged by a third type of strategy, the DH model (**Greulich and Simons, 2016**). Motivated by the emerging view of prevailing cell plasticity (**Blanpain and Fuchs, 2014**; **Tetteh et al., 2015**; **Tetteh et al., 2016**; **Donati and Watt, 2015**), the DH model considers the possibility of reversible switching between two cell types:

$$S \xrightarrow{\lambda} S+D, \quad S \underset{\omega_S}{\overset{\omega_D}{\rightleftharpoons}} D, \quad D \xrightarrow{\gamma} \emptyset. \tag{2}$$

where symbols at arrows denote the *process rates* (frequency of events). This strategy is also capable of maintaining a homeostatic population if $\gamma/\lambda = \omega_S/\omega_D$. Notably, the DH model only features asymmetric divisions (in that daughter cells are of different type), like the IA model, yet the DH model predicts clonal statistics that are indistinguishable from the PA model (**Greulich and Simons, 2016**). This means that in view of the existing clonal data for mouse tissues, the DH model, may as well describe the real cell fate dynamics. More fundamentally, this implies that the PA and DH model cannot be distinguished via plain clonal data, which poses fundamental limitations to the common approach to use lineage tracing for determining cell fate choices.

This demonstrates that the classical definition of asymmetric and symmetric divisions is not always suitable to distinguish cell fate strategies in view of clonal data alone. In general, cell fate dynamics may be much more complex than the simplified models described above, as there may exists a plethora of cell (sub-)types in a tissue. However, to what extent would it be possible to distinguish details of potentially rather complex cell fate dynamics models through comparison with clonal data at all? This is only the case if the clonal statistics are sufficiently different. In the following, we study cell fate models in their most generic form, and analyze what clonal statistics would be expected.

## Results

### Model generalization

Let us consider the dynamics of a generic system of cells, characterized by a number $m$ of possible cell states $X_i$, $i = 1, ..., m$. We define a cell *state* here as a group of cells showing common properties (e.g. any cell sub-type classification). Most generally, cells in a state $X_i$ may be able to divide, producing daughter cells of any cell states $X_j$ and $X_k$ (where $i = j = k$, that is, simple cell duplication, is possible). Furthermore, any cell state $X_i$ may turn into another state $X_j$ or may be lost (through emigration, shedding, or death). Hence, we can write a generic cell fate model as,

$$\text{cell division:} \quad X_i \xrightarrow{\lambda_i r_i^{jk}} X_j + X_k \tag{3}$$

$$\text{cell state change:} \quad X_i \xrightarrow{\omega_{ij}} X_j \tag{4}$$

$$\text{cell loss:} \quad X_i \xrightarrow{\gamma_i} \emptyset, \tag{5}$$

where $i, j, k = 1, ..., m$. In this model, $\lambda_i$ is the rate of division of cells in state $X_i$ and the parameter $r_i^{jk}$ corresponds to the proportion of division outcomes producing daughter cells of state $X_j$ and state $X_k$; $\omega_{ij}$ is the transition rate from state $X_i$ to state $X_j$ and $\gamma_i$ the loss rate from state $X_i$.

The dynamics of each cell in *Equations 3-5* could depend on the cell environment through spatial, cell-extrinsic regulation of cell fate. However, the clonal statistics of spatial models that include

cell-extrinsic regulation of cell fate (models of the voter type [**Clifford and Sudbury, 1973**]) are, in the long term, the same as for the corresponding branching process models (**Haccou et al., 2005**), as Equations 3-5 are, except for one-dimensional arrangements of cells (as shown in **Klein and Simons, 2011**; **Bramson and Griffeath, 1980**). Here, we are focussing on the long-term clonal statistics of self-renewal strategies, and since this is not affected by cell-extrinsic regulation, for tissues with two-dimensional or three-dimensional arrangements of dividing cells (like epithelial sheets, and volumnar tissue), we wish to keep the analysis simple and therefore choose dynamics (and thus the parameters $\lambda_i, \omega_{ij}, r_i^{jk}, \gamma_i$) to be independent of the cell environment.

In the following, we study the dynamics of cell numbers in each state $X_i$, $n_i$. To gain initial insight into those dynamics, let us first consider the time evolution of the *mean* cell numbers, $\bar{n}_i = \langle n_i \rangle$, given by,

$$\frac{d}{dt}\bar{n}_i = \sum_j \left(\lambda_j 2r_j^i + \omega_{ji}\right)\bar{n}_j - (\lambda_i + \sum_j \omega_{ij} + \gamma_i)\bar{n}_i . \tag{6}$$

in which $r_i^j = \sum_k (r_i^{jk} + r_i^{kj})/2$ is the probability of having a daughter cell in state $X_j$ produced upon division of a cell in state $X_i$. This linear system of differential equations can be written more compactly in terms of the mean cell number vector $\bar{n} = (\bar{n}_1, \bar{n}_2, ..., \bar{n}_m)$,

$$\frac{d}{dt}\bar{n} = A\bar{n}, \tag{7}$$

with $A$ being the $m \times m$ matrix

$$A = \begin{pmatrix} \kappa_{11} - \delta_1 & \kappa_{21} & \kappa_{31} & \cdots \\ \kappa_{12} & \kappa_{22} - \delta_2 & \kappa_{32} & \cdots \\ \kappa_{1m} & \kappa_{2m} & \cdots & \kappa_{mm} - \delta_m \end{pmatrix}, \tag{8}$$

where we defined the *total transition rate* $\kappa_{ij} = \lambda_i 2r_i^j + \omega_{ij}$, combining all transitions from $X_i$ to $X_j$ by cell divisions and direct transitions, and the *local loss rate* $\delta_i = \lambda_i + \sum_j \omega_{ij} + \gamma_i$.

Models of the form **Equations 3–5** are not generally in homeostasis, which in this context is defined by the existence of a stationary state $\bar{n}^*$, with $d\bar{n}^*/dt = 0$, that is (Lyapunov) stable and non-trivial (for a discussion, see the Appendix 'Conditions for homeostasis'). This can in principle be assessed through the spectral properties of $A$ (**Åström and Murray, 2008**), but applying spectral conditions explicitly is unwieldy and difficult to interpret biologically. For a more intuitive view, we interpret the system, **Equation 7**, as a network (graph): the matrix $A$ can be interpreted as the adjacency matrix of the *cell state network*. This is a weighted directed graph in which cell states correspond to the graph's nodes and a link from state $X_i$ to $X_j$ exists where a transition is possible, that is, when $\kappa_{ij} > 0$. The value of $\kappa_{ij}$ also denotes the link weights (diagonal elements of $A$ can be considered as self-links). Now, we note that **Equation 7** is linear and cooperative, that is, the off-diagonal elements of matrix $A$ are non-negative, and for such systems more simple and intuitive conditions for homeostasis exist (**Greulich et al., 2019**), based on a decomposition into the network's *Strongly Connected Component (SCC)*. An SCC is a sub-graph that groups nodes which are *strongly connected*, that is, which are mutually connected by paths (more accurately: two nodes, $X_i$ and $X_j$, are strongly connected if there exists a path from $X_i$ to $X_j$ and from $X_j$ to $X_i$ on the network). An example of such a decomposition, which yields an *acyclic* condensed network that contains SCCs as nodes and directed links between them, is shown in **Figure 1**.

The stability of systems like **Equation 7** is then determined by the dominant eigenvalues $\mu_k$ of each strongly connected component $k$, for $k = 1, ..., m_S$ where $m_S$ is the number of SCCs, and their topological arrangement (the Perron-Frobenius theorem assures that for adjacency matrices of SCCs of cooperative systems, a unique, real, maximal eigenvalue exists, which is the dominant eigenvalue [**Arrow, 1989**; **Greulich et al., 2019**]). In brief, according to **Greulich et al., 2019**, the conditions for existence of a homeostatic state are that, at the apex of each lineage (the condensed cell state network), there must be an SCC with dominant eigenvalue $\mu_k = 0$, while all SCCs downstream of the former must have $\mu_k < 0$ (see detailed discussion in the Appendix, 'Conditions for homeostasis'). Given this structure of homeostatic models, we can define two compartments in the cell state transition network: (1) the (self-) Renewing compartment ($\mathcal{R}$), which is the SCC at the apex of the lineage tree;

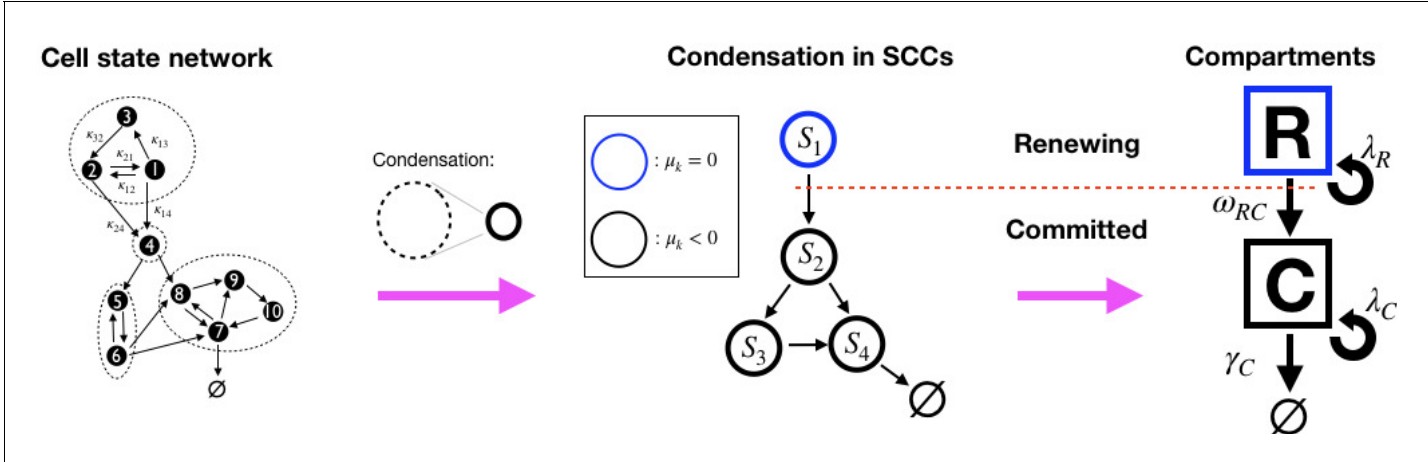

**Figure 1.** Illustration of the decomposition of a homeostatic cell state network into SCCs and the compartment representation, *Equation 9*. (Left): An example cell state network representing the matrix $A$ in *Equation 8* (self-links not displayed). The dashed circles denote the network's Strongly Connected Components (SCCs ) (see definition in text). (Middle): The *Condensed network* is the corresponding network of SCCs, $S_k$, wherein SCCs are the nodes and a link between two SCCs exists if any of their states are connected. For homeostatic networks, an SCC with dominant eigenvalue $\mu = 0$ is at the apex, while other SCCs have $\mu < 0$. (Right): We distinguish two compartments, the Renewing compartment $\mathcal{R}$, consisting of the apex SCC, with $\mu = 0$, and the Committed compartment $\mathcal{C}$ consisting of the remainder, with $\mu < 0$.

and (2) the Committed compartment ($\mathcal{C}$), which consists of all SCCs with $\mu_k < 0$, that is, those downstream of the apex SCC. Importantly, cells in states forming $\mathcal{R}$ have the potential to return to any state within the same compartment and this population maintains itself. Instead, the cell population in $\mathcal{C}$ would vanish without external input, since the combined dominant eigenvalue of all those SCCs is negative (it is the maximum of all SCCs' $\mu_k < 0$), thus the progeny of each cell in the committed compartment will eventually be lost. We can thereby classify cells as being of a (self-)*Renewing type* (R) if their state is within $\mathcal{R}$, and of a *Committed type* (C) if their state is in $\mathcal{C}$. With this coarse-grained classification, a generic homeostatic model can be represented in terms of compartments $\mathcal{R}$ and $\mathcal{C}$ as,

$$R \xrightarrow{\lambda_R} \begin{cases} R+R & \text{with probability } r_{RR} \\ R+C & \text{with probability } 1 - r_{RR} - r_{CC}, \\ C+C & \text{with probability } r_{CC} \end{cases} \tag{9}$$

$$R \xrightarrow{\omega_{RC}} C, \quad C \xrightarrow{\lambda_C} C+C, \quad C \xrightarrow{\gamma_C} \emptyset,$$

where the symbols above arrows are the *effective rates* of those events, denoting the average frequency at which they occur (loss events $R \to \emptyset$ are not explicitly included, since they can be approximated by a short lived state $X_d$ in $\mathcal{C}$, as $R \to X_d \to \emptyset$). To be compatible with a homeostatic condition, it is further required that (i) the R-population remains on average constant ($\mu_k = 0$), that is, $\lambda_R r_{RR} = \lambda_R r_{CC} + \omega$, and (ii) the loss rate of C must exceed its proliferation rate ($\mu_k < 0$), that is, $\gamma_C > \lambda_C$. *Figure 1* shows how a generic homeostatic cell state network can be condensed into an effective model of renewing and committed cell states, according to *Equation 9*. It has to be noted, however, that the events depicted in *Equation 9* are *not Markovian*, that is, the timing of events is not independent from each other and depends on their history. Thus, the 'rates' $\lambda_R$, $\lambda_C$, $\omega_{RC}$, and $\gamma_C$ are not constant rates in the Markovian sense, yet we can define them by the mean frequency of events occurring (see Appendix 'Approximation of generic GIA models' and 'Asymptotic clone size distributions: mathematical analysis').

The formulation in terms of renewing and committed states can help us to gain insights into potential behaviors of generic homeostatic cell fate models. In particular, we define *generalized asymmetric divisions* as events of the type $R \to R + C$, and *generalized symmetric divisions* as events of the type $R \to R + R$ (symmetric renewal) and $R \to C + C$ (symmetric commitment). With these

definitions, we can categorize homeostatic cell fate models into two classes: *Generalized Invariant Asymmetry* (GIA) models are those which only exhibit $R \rightarrow R + C$ divisions in the renewing compartment, while *Generalized Population Asymmetry* (GPA) are models for which such restriction does not hold. We note that the two classes are equivalently characterized by a conservation law: For GIA models, the number of cells in $\mathcal{R}$ is strictly conserved, while for GPA models, no such conservation law holds. Since $\mu = 0$ is necessary for conservation, the only possible conserved cell states in homeostasis are those in $\mathcal{R}$. Naturally, the previously discussed IA model is a GIA model and the PA model is a GPA model. Notably, the DH model (*Equation 2*) is of the GPA category, since in that model $S$ and $D$ cells form a single SCC at the apex of the lineage hierarchy, and thus they are both part of $\mathcal{R}$. Therefore, a division $S \rightarrow S + D$ in the DH model, which is asymmetric in the conventional sense, corresponds to $R \rightarrow R + R$ in terms of compartments (*Equation 9*) and thus it is a generalised symmetric division. According to this classification, PA and DH models are both in the same category (GPA), and indeed, both predict the same type of clone size distribution, an Exponential one (*Greulich and Simons, 2016*).

## Numerical simulation of random cell fate models

To check whether the correspondence between model class, GIA vs. GPA, and predicted clonal statistics holds in general, we analyze the clonal dynamics numerically, by generating and testing a large number of random stochastic models, implemented via random generation of the parameters $\lambda_i$, $\omega_{ij}$, $\gamma_i$ and $r_i^{jk}$. To simulate clones, we perform stochastic simulations based on the Gillespie algorithm (*Gillespie, 1977*), assuming a Markov process following the rules of *Equation 3-5*. We run, for each model, a large number of simulations with initially one cell in the compartment $\mathcal{R}$, thus the cell population of each simulation run represents one clone. Then we sample their outcomes, the total cell numbers per clone (the *clone size*) $n = \sum_i n_i$, to obtain predictions for clonal statistics, namely the frequency distribution of clone sizes (*clone size distribution*) and mean clone sizes (see Materials and methods).

We first study the mean clone size of surviving clones (with $n>0$), $\bar{n}_s = \langle n \rangle|_{n>0}$, shown in *Figure 2*, respectively, for the GIA and GPA models, as a function of time (the final time $\tau = 20/\alpha_{\min}$ where $\alpha_{\min}$ is the minimal process rate, $\alpha_{\min} = \min(\lambda_1, ..., \omega_{12}, ..., d_m)$). We note that indeed a common behavior is seen in each case. While for every simulated GIA model, $\bar{n}_s$ saturates at a plateau value, it steadily increases for every GPA model. This is expected, and can be understood given that clones in a GPA model can go extinct while those in a GIA model not. Assume that there are initially a large number $N_c$ of clones, such that the total number of cells is $n_{\text{tot}} = N_c \bar{n}_s$. Since the system is homeostatic, it will reach a constant steady state $n_{\text{tot}}^*$ after a sufficient amount of time, meaning that the mean clone size is $\bar{n}_s = n_{\text{tot}}^*/N_c$. If no clones go extinct, as in GIA models, $N_c$ is constant and thus $\bar{n}_s$ approaches a constant. However, in non-conserved multi-type branching processes, as GPA models are, the clone number $N_c$ decreases through progressive extinction of clones (*Haccou et al., 2005*), and therefore $\bar{n}_s$ increases, despite the cell population as a whole staying stationary.

The resulting clone size distributions for the two model classes are shown in *Figure 3*. Here, clones sizes $n$ are rescaled by the mean value $\bar{n}_s$ and compared to an Exponential distribution of unitary mean (green curve). As conjectured, all simulated GPA models shown in panel (b) predict asymptotically the same rescaled clone size distribution, namely a standard Exponential distribution. Deviations exist for small times and small clone sizes, but these deviations vanish in the large time limit (details on the convergence are shown in the Appendix, 'Analysis of the generalized Population Asymmetry model'). This means that different models within the GPA class cannot be distinguished in the long-term limit, since they differ only by the mean clone size, which is a free fit parameter. In analogy to statistical physics, we can categorize them as a *universality class* (*Klein and Simons, 2011*), meaning that the details of the model do not affect the (scaled) outcomes for assymptotic conditions, which is a form of weak convergence of random variables (*Billingsley, 1968*). However, the same cannot be said about the GIA models. In fact, we see all kind of shapes in the clone size distributions, both peaked distributions and non-peaked ones, and in fact, some distributions are even close to an Exponential form, and can thus not be distinguished from GPA models. The question is whether we can yet find other parameters for which, when large, also GIA models exhibit universality, that is, yield the same rescaled clone size distribution. For this purpose, we will in the following sections develop a deeper theoretical understanding of the model classes.

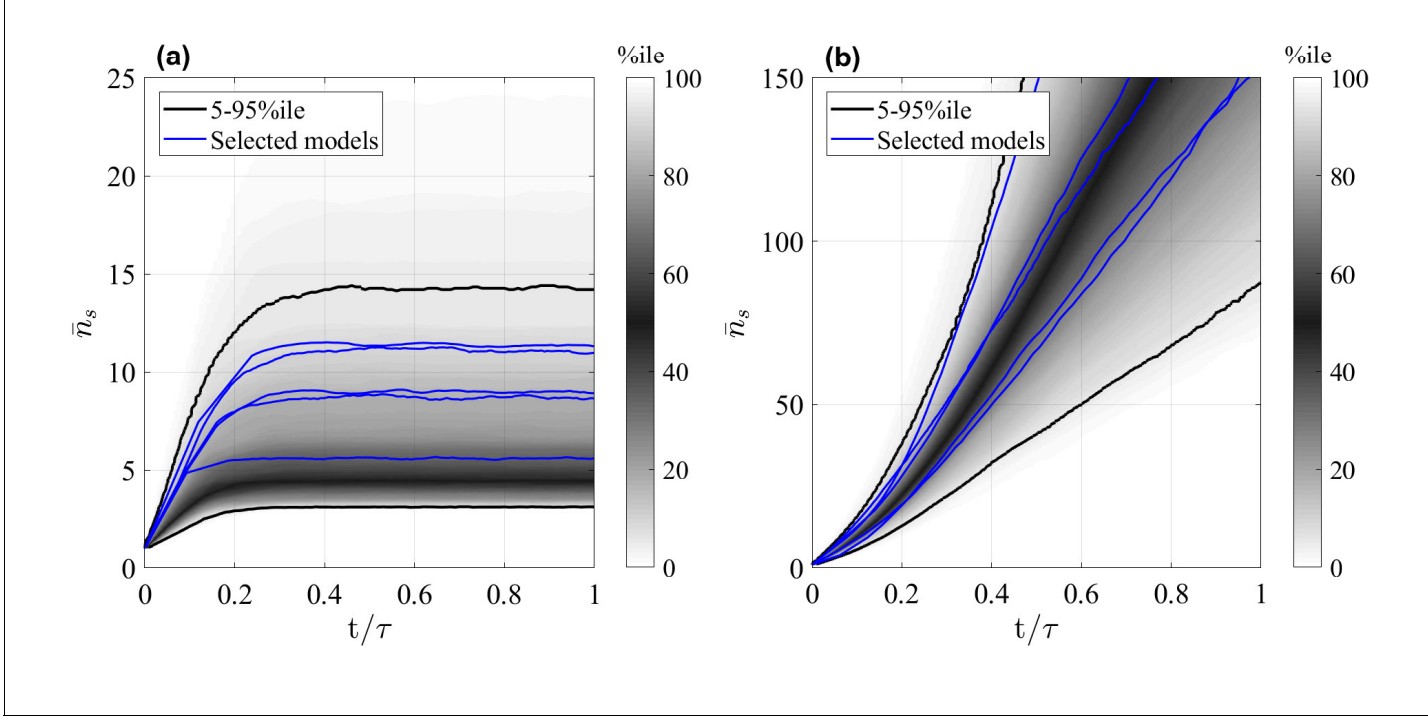

**Figure 2.** Mean size of surviving clones, $\bar{n}_s$, as a function of time for random GIA models (a), and GPA models (b). In (a), $\tau = 20/\alpha_{\min}$, in (b), $\tau$ is the time at 98% clone extinction. The grey shade represents the percentile of all the simulations (black lines limit the 5-95%ile range); the blue curves correspond to some illustrative selected simulations. Simulations for which the final mean is below two and where the final condition is not achieved (due to computational limitations) are not included: this results in 238 and 571 models, respectively for the GIA and GPA cases.

## Mathematical analysis: Markovian approximation of compartment model

To obtain a deeper understanding of the numerical results, we study the cell fate models in terms of the compartment representation, *Equation 9*. In this representation models are not Markovian, yet we can study their Markovian counterpart, as an approximation. While this is not expected to yield accurate clone size distributions in general, the limiting distributions of non-Markovian processes are commonly well estimated by their Markovian counterparts.

For GIA models, which only feature $R \to R + C$ transitions between the renewing compartment, $\mathcal{C}$, and the committed compartment, $\mathcal{C}$, a corresponding Markovian model reads,

$$X_1 \xrightarrow{\lambda_1} X_1 + X_2, \qquad X_2 \xrightarrow{\lambda_2} X_2 + X_2, \qquad X_2 \xrightarrow{\gamma} \emptyset, \tag{10}$$

in which $X_1$ represents a single state in $\mathcal{R}$ and $X_2$ in $\mathcal{C}$, and symbols at arrows are the process rates. The number of cells in $X_1$, $n_1$, is conserved, that is, given an single $X_1$-cell initially, it always remains at $n_1 = 1$. Thus, we only need to consider the dynamics of cells in $X_2$, $n_2$. This Markov process can be solved analytically, and for sufficiently large steady state mean number of $X_2$-cells, $\bar{n}_2 = \langle n_2 \rangle = \lambda_1/(\gamma - \lambda_2)$ (see Appendix, 'GIA$^0$ test case: steady state distribution and limiting behavior'), the rescaled distribution of cells in $X_2$ is,

$$P(x_2) = (1 - \hat{\lambda}_2)^{\frac{\hat{\lambda}_1}{\hat{\lambda}_2}} \hat{\lambda}_2^{\frac{\hat{\lambda}_1 x_2}{(1 - \hat{\lambda}_2)}} \frac{\Gamma\left(\frac{\hat{\lambda}_1}{\hat{\lambda}_2} + \frac{\hat{\lambda}_1}{1 - \hat{\lambda}_2} x_2\right)}{x_2 \Gamma\left(\frac{\hat{\lambda}_1}{\hat{\lambda}_2}\right) \Gamma\left(\frac{\hat{\lambda}_1}{1 - \hat{\lambda}_2} x_2\right)}, \tag{11}$$

in which $x_2 = n_2/\bar{n}_2$, $\hat{\lambda}_1 = \lambda_1/\gamma$ and $\hat{\lambda}_2 = \lambda_2/\gamma$ and $\Gamma(...)$ is the Gamma function (*Abramowitz and Stegun, 1972*). We note that this distribution exhibits a large variety of shapes: for large $\hat{\lambda}_1$ the distribution is peaked, while for small $\hat{\lambda}_1$ is loses its peak. Notably, for $\hat{\lambda}_1 \to 1$ and $\hat{\lambda}_2 \to 1$, the distribution becomes Exponential and in this case it cannot be distinguished from the GPA case. On the other hand, for $\hat{\lambda}_1 \to \infty$, that is, when the ratio of asymmetric divisions over the loss rate is high, this

distribution tends to a Normal distribution with unitary mean and variance equal to $1/\hat{\lambda}_1$. These different behaviors are graphically shown in the Appendix (see *Appendix 1—figure 6*, *7* and *8*).

For the GPA models, a Markovian approximation reads, accordingly,

$$X_1 \xrightarrow{\lambda_1} \begin{cases} X_1 + X_1 & \text{with probability } r_1 \\ X_1 + X_2 & \text{with probability } 1 - r_1 - r_2, \\ X_2 + X_2 & \text{with probability } r_2 \end{cases} \tag{12}$$

$$X_1 \xrightarrow{\omega} X_2, \qquad X_2 \xrightarrow{\lambda_2} X_2 + X_2, \qquad X_2 \xrightarrow{\gamma} \emptyset.$$

whereby for homeostasis to prevail, $\lambda_1 r_1 = \lambda_1 r_2 + \omega$ and $\lambda_2 < \gamma$ must hold. We note that the dynamics of $X_1$ are independent of $X_2$ and thus for the number of cells in $X_1$ in homeostasis holds

$$n_1 \xrightarrow{\lambda_1 r_1 n_1} n_1 \pm 1, \tag{13}$$

which corresponds to a simple continuous-time branching process with two offspring, for which it is known that the resulting distribution of cell numbers is Exponential, that is, $P_1(n_1) = \bar{n}_{1,s}^{-1} e^{-n_1/\bar{n}_{1,s}}$, where $\bar{n}_{1,s} \simeq \lambda_1 r_1 t$ is the mean number of cells in the surviving clones (*Haccou et al., 2005*).

$X_2$ cells produced according to 12 follow the same fate as in the two-state GIA model above. While it is not assured that the distribution of $X_2$ cells is identical to that of *Equation 11* (due to simultaneous production events of type $X_1 \rightarrow X_2 + X_2$), we show in the Appendix, 'Asymptotic clone size distributions: mathematical analysis', that for large rates of production of C-cells, the distribution of C-cells – here: cells in state $X_2$ – attains a Normal distribution with mean $\bar{n}_2$ equal to its variance $\sigma_{n_2}^2 = \langle (n_2 - \bar{n}_2)^2 \rangle = \bar{n}_2$. As each $X_1$ cell contributes independently to the production of $X_2$-cells, we have that $\bar{n}_2 \sim n_{1,s} \sim t$. Crucially, this means that in terms of the rescaled variable $x_2 = n_2/\bar{n}_s$ the standard deviation $\sigma_{x_2} = \frac{\sigma_{n_2}}{\bar{n}_s} \leq \frac{1}{\sqrt{\bar{n}_2}} \sim t^{-1/2}$ vanishes for large times, since $\bar{n}_2 \sim n_{1,s} \sim t \rightarrow \infty$. Hence, given fixed $x_1$, $x_2$ can be approximated by a constant random number $x_2|_{x_1} \sim \bar{x}_1 = n_1/\bar{n}_s$. Therefore, the

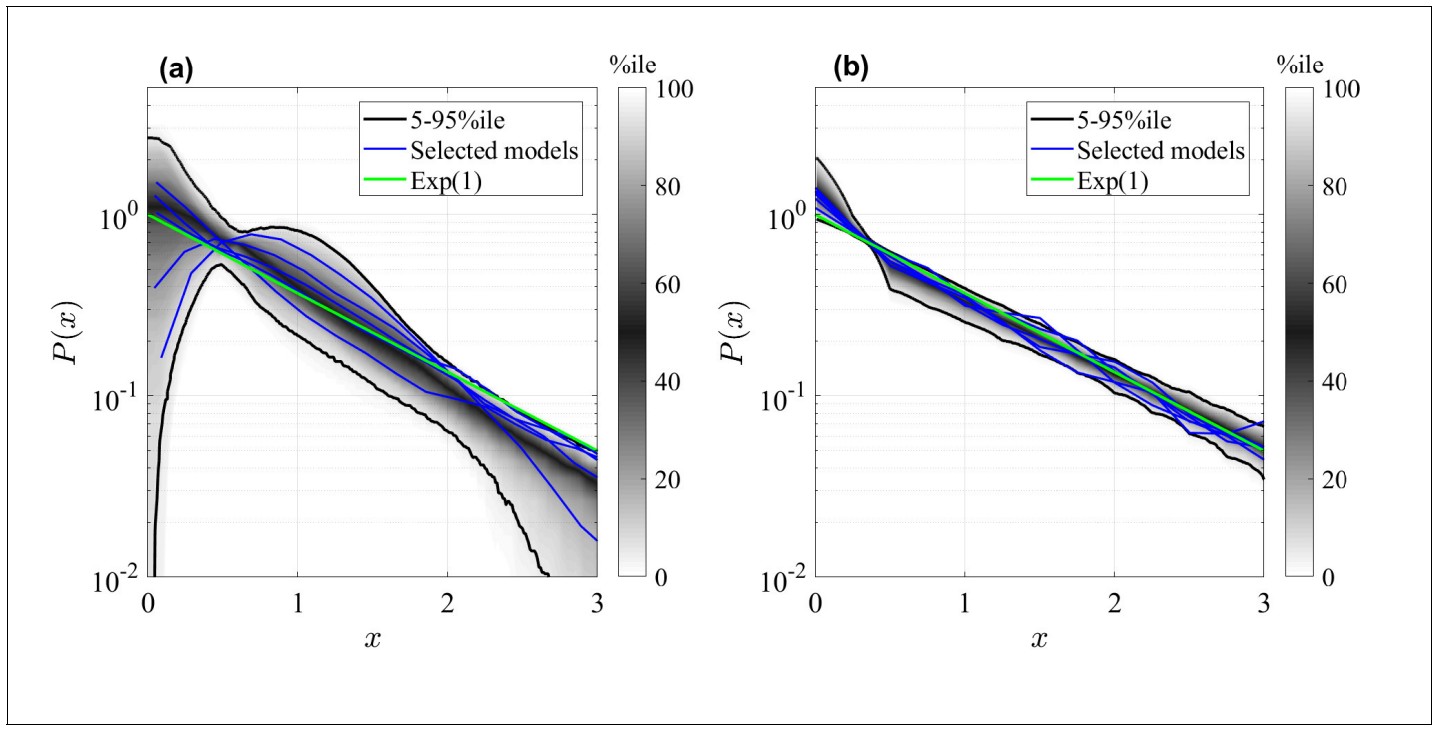

**Figure 3.** Rescaled clone size distributions (expected relative frequency *P* of clone sizes) for random GIA models (a), and GPA models (b), in terms of the rescaled clone size $x = n/\bar{n}_s$, at final time $t = \tau$ (see *Figure 2* for definition). The grey shade represents the percentile of all the simulations (black lines limit the 5-95%ile range); the blue curves correspond to some selected simulations. A reference curve corresponding to an Exponential distribution of unitary mean ('Exp(1)') is shown in green.

rescaled distribution of the total number of cells is $P(x) = P_1(x - x_2) = e^{-x}$, where $\bar{x} = \bar{x}_1 + \bar{x}_2 \sim \bar{x}_1$. Thus, the rescaled distribution of the total clone size, $x = n/\bar{n}_s$, is as well an Exponential.

## Universality of generic cell fate models

For generic GIA or GPA models, the compartment representation, *Equation 9*, is not Markovian and one would not expect exactly the distributions we found in the previous section. Fortunately, the limiting distributions of non-Markovian processes and their Markovian counterparts are often, under certain conditions on the parameters, the same. While we reserve the technical arguments for the Appendix ('Asymptotic clone size distributions: mathematical analysis'), we note that this independence of the limiting distribution on the Markov property related to the central limit theorem, which does not rely on the Markov property.

To identify the correct limiting parameters for more complex cell fate models, we need to express the effective non-Markovian rates (i.e. the mean frequency of events) of representation nine in terms of the original model, 3–5. As discussed in the Appendix ('Approximation of generic GIA models' and 'Asymptotic clone size distributions: Mathematical analysis'), we identify those effective rates by the total rates of cell divisions, $\lambda_R = \sum_{i \in \mathcal{R}} \lambda_i P_i^R$, $\gamma_C = \sum_{i \in \mathcal{C}} \gamma_i P_i^C$, and $\omega_{RC} = \sum_{i \in \mathcal{R}, j \in \mathcal{C}} \omega_{ij} P_i^R$ where, for each compartment, $P_i^{R,C} = \bar{n}_i / \sum_{j \in \mathcal{R}, \mathcal{C}} \bar{n}_j$ is the probability of a single cell being in state $X_i$ of $\mathcal{R}$, respectively ($\bar{n}_i$ are the solutions to *Equation 6*). In the Appendix, 'Asymptotic clone size distributions: mathematical analysis', we reason that all GPA models are expected to generate Exponential clone size distributions for large times $t$. This is indeed what is observed in *Figure 3(b)*. Correspondingly, for GIA models we expect that for large $\hat{\lambda}_R = \lambda_R / \gamma_C$ the clone size distribution of GIA models would tend to a Normal distribution. To test this prediction, we simulated the same GIA models as for *Figure 3* before, but we tuned parameters in $\mathcal{R}$ such that the effective parameter $\hat{\lambda}_R$ becomes large (see details in the Appendix, 'GIA model for large $\hat{\lambda}_R$'). The result is shown in *Figure 4*: for an illustrative case shown in panel (a), increasing $\hat{\lambda}_R$ changes the distribution from an exponential form to a peaked form akin to a Normal distribution, and for all simulated random GIA models, shown in panel (b), a Normal distribution is approached when $\hat{\lambda}_R$ becomes large.

We note that when taking the limit of large $\hat{\lambda}_R$, as shown in *Figure 4*, also all other process rates $\omega_{ij}$ with $i,j$ within $\mathcal{R}$ increased as well. What if instead some process rates in $\mathcal{R}$ do not scale to become large with $\hat{\lambda}_R$? To assess this situation, we studied a simple test case similar to model 10 but containing two states in $\mathcal{R}$, connected via direct state transition (see Appendix, 'GIA$^B$ test case: bimodal distribution'). As discussed there, if all rates within $\mathcal{R}$ are large compared to the rates in $\mathcal{C}$ then indeed we observe a Normal clone size distribution, as expected. However, if the direct transition rates between the states of $\mathcal{R}$ are smaller or of equal magnitude as $\gamma_C$, and in addition, one of the two division rates is higher then the other, then we observe a bimodal clone size distribution. The reason is that if the transitions between the two states in $\mathcal{R}$ are rare compared to the life time of cells, $1/\gamma_C$, they become essentially separated and each of those states generate separate Normal distributions with different mean (due to different cell division rates in those two states) which, when overlaid, generate a bimodal clone size distribution (see detailed arguments in the Appendix, 'Asymptotic clone size distributions: mathematical analysis').

Finally, from those considerations follows:

1. GPA models attain an Exponential clone size distribution for time $t \to \infty$.
2. GIA models attain a Normal clone size distribution if all process rates within $\mathcal{R}$ are much larger than the inverse lifetime of $C$-cells, $\gamma_C$.

Hence, the GIA and GPA model classes, each represent a universality class, that is, a scaling limit exists in which all models of the same class yield the same rescaled clonal statistics.

## Discussion

Our analysis shows that intrinsic limitations exist for identifying strategies of stem cell self-renewal through clonal data from cell lineage-tracing experiments. This is due to different models of cell fate choice generating the same type of clonal statistics (clone size distributions), so that model inference based on clonal statistics – currently still the most prevalent method to determine stem cell self-renewal strategies – fails to distinguish them. The feature that different models asymptotically

generate the same statistics is a form of weak convergence of random variables (**Billingsley, 1968**) and corresponds to *universality*, as known from statistical physics.

Cell fate models can in principle be very complex, with a plethora of cell (sub-)types in a tissue. We introduced a new categorization of cell types, distinguishing between cell states that are committed (*C*-cells), whose progeny is inevitably lost eventually, and non-committed or (self-)renewing cell states (*R*-cells), which retain the potential to remain or return to the apex of the lineage hierarchy. According to this categorization we classified generic models of cell fate choice as *Generalized Invariant Asymmetry* (GIA), if only generalized asymmetric divisions of the form $R \rightarrow R + C$ occur for *R*-cells, and *Generalized Population Asymmetry* (GPA), when all kind of divisions can occur, as long as gain and loss of *R*-cells are balanced. Models of the GIA category are also characterized by a conservation law, since the number of *R*-cells is strictly conserved, while GPA models do not exhibit such a conservation law.

We found that the classification in GIA and GPA models mirrors the clonal statistics generated by them: models of the GPA class all generate clonal statistics which with time converge to an Exponential clone size distribution. Thus, two GPA models can therefore not be distinguished through clonal data, once some time has passed after induction of clones. For GIA models, distributions can generally vary, but if the rates of divisions and transitions in the $\mathcal{R}$ compartment are much larger that the rate of cell loss, the clone size distribution of all those models becomes a Normal distribution. In that case, two GIA models can not be distinguished by the clonal data. While here we do not explicitly consider cell-extrinsic regulation of cell fate, this kind of regulation does not affect long-term clone size distributions, except when cells are arranged one-dimensionally (**Klein and Simons, 2011**; **Bramson and Griffeath, 1980**). Thus, our results cover cell dynamics in most renewing tissues, such as epithelial sheets or volumnar organs, but not (quasi-)one-dimensional arrangements of stem cells, as found in the seminiferous tubule, or in intestinal crypts, where clonal statistics may differ. Hence, our analysis shows that models of cell fate choice cannot in general be distinguished with further resolution beyond the *R* vs. *C* categorization of cell types. The universality of the model dynamics also

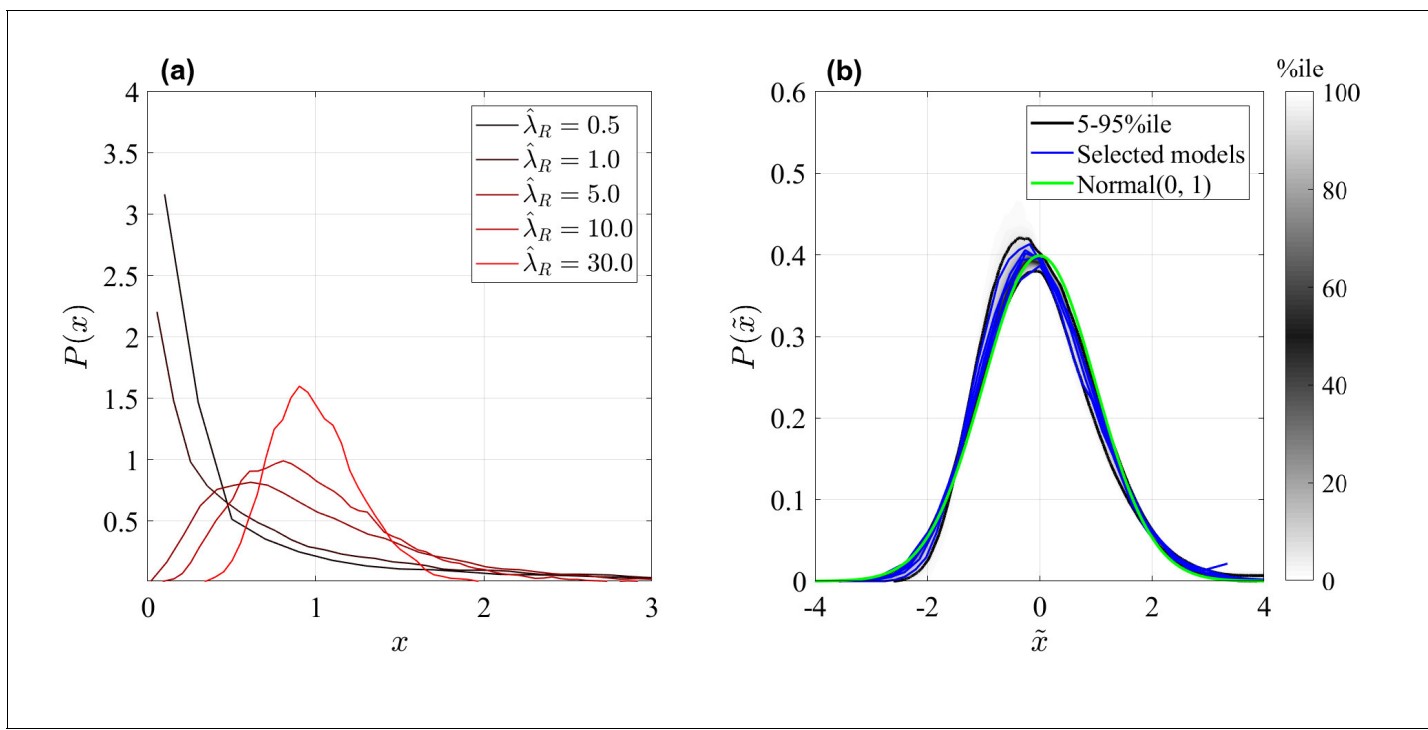

**Figure 4.** Rescaled clone size distributions (expected relative frequency *P* of clone sizes) for random GIA models as in **Figure 3**, at time $t = \tau$ (see definition in **Figure 2**). Sensitivity to parameter $\hat{\lambda}_R$ is shown for one illustrative case in panel (a), and all GIA models for $\hat{\lambda}_R = 30$ in panel (b). The distributions are shown in terms of the rescaled variables $x = n/\bar{n}_s$ for panel (a) and $\tilde{x} = (n - \bar{n}_s)/\sigma_n$, where $\sigma_n$ is the distributions variance, in panel (b). In (b), the grey shade represents the percentile of all simulations (black lines limit the 5-95%ile range); the blue curves correspond to some selected simulations. A reference curve corresponding to a Normal distribution of zero mean and unitary variance is shown in green. Simulations for which $t = \tau$ is not reached (due to computational limitations) are not included, resulting in 922 model instances.

shows that effective, simplistic models are often equally accurate to model experimental data, yet with a higher statistical power due to less free parameters.

While at first glance, this analysis seems to discourage efforts to unravel details of cell fate dynamics, room remains in regimes where the limiting conditions for asymptotic distributions are not fulfilled. In particular, if fast cycling committed progenitor cells are present, while stem cells are slow cycling, then the condition that the division rate of *R*-cells is much larger than the cell loss rate is not fulfilled. In that case, details of the model dynamics may affect the shape of the clone size distribution and thus allow distinction between models. However, caution should be given when an Exponential clone size distribution is observed, since this could indicate either a GIA model with high activity of committed progenitor cells, or a GPA model. In that case, the mean clone size needs to be consulted to distinguish models (see *Figure 2*). Differentiating between models within the GPA category is more difficult, since the predicted statistics from different models always become more similar over time. Short-term measurements would in principle allow such a distinction, but since in reality the underlying processes are not truly Markovian (as assumed for the modeling purpose) they are not necessarily a good representation of the real cell dynamics at short times. At long times, however, Markovian approximations are increasingly accurate, precisely because of the feature of universality.

How could the resolution of cell fate modeling be improved? The state-of-the-art approach to determine cell fate trajectories is via analysis and modeling of single-cell RNA-sequencing (scRNA-seq) data. However, many limitations to this method exist, discussed in *Weinreb et al., 2018*, and neither reversible trajectories nor the modes of cell division, such as asymmetric vs symmetric divisions, can be inferred. Intravital live imaging, on the other hand, allows to trace individual clones over time (*Ritsma et al., 2012*; *Pittet and Weissleder, 2011*; *Hara et al., 2014*; *Rompolas et al., 2016*), and thus can obtain details of cell fate trajectories, yet this technique is limited to few tissue types which are accessible for invasive long-term imaging. Nonetheless, while each of those experimental assays alone is prone to limitations in defining self-renewal strategies, advanced model inference schemes, that integrate data from different experimental sources, might be the way forward in the future to finally reveal the details of stem cell self-renewal strategies.

## Materials and methods

The numerical analysis of the random cell fate model was implemented in Matlab. The description of the stochastic models definition, the random model generation and the simulation campaign is detailed in the Appendix, 'Stochastic process modelling'. Additionally, as a validation of the implemented simulator, based on the Gillespie algorithm (*Gillespie, 1977*), the IA and PA models were simulated and the results analyzed in the Appendix, 'Invariant Asymmetry and Population Asymmetry models'.

Analytical solutions were partially obtained using Mathematica.

## Acknowledgements

We thank Benjamin D MacArthur for valuable discussions that contributed in the development of this research. CP is supported by a Studentship of the Institute for Life Sciences (Southampton) and PG by Medical Research Council New Investigator Research Grant MR/R026610/1.

## Additional information

### Funding

| Funder | Grant reference number | Author |
| --- | --- | --- |
| Medical Research Council | MR/R026610/1 | Philip Greulich |

The funders had no role in study design, data collection and interpretation, or the decision to submit the work for publication.

## Author contributions
Cristina Parigini, Conceptualization, Software, Formal analysis, Validation, Investigation, Visualization, Methodology, Writing - original draft, Writing - review and editing, Mathematical analysis (part), Numerical analysis; Philip Greulich, Conceptualization, Supervision, Funding acquisition, Project administration, Writing - review and editing, Mathematical analysis (part)

## Author ORCIDs
Philip Greulich (iD) https://orcid.org/0000-0001-5247-6738

## Decision letter and Author response
Decision letter https://doi.org/10.7554/eLife.56532.sa1
Author response https://doi.org/10.7554/eLife.56532.sa2

## Additional files
### Supplementary files
• Transparent reporting form

### Data availability
All numerical data used for figures is produced by programme code, which can be found on Github, under https://github.com/cp4u17/simCellState (copy archived at https://github.com/elifesciences-publications/simCellState).

The following dataset was generated:

| Author(s) | Year | Dataset title | Dataset URL | Database and Identifier |
|---|---|---|---|---|
| Parigini C | 2020 | simCellState | https://github.com/cp4u17/simCellState | Github, cp4u17/simCellState |

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

## Appendix 1

### Conditions for homeostasis

Here, we 'translate' the generic conditions for the existence of a Lyapunov stable stationary state for Linear Cooperative Systems (*LCS*) (*Greulich et al., 2019*) into the biological context of clonal dynamics. A linear cooperative system is one of the form $\frac{d}{dt}x(t) = Ax(t)$ where $x(t) = (x_1(t), x_2(t), ..., x_m(t))$ are functions of time t and A is a constant m × m matrix for which all off-diagonal elements are non-negative (the latter condition defines the cooperativity of the system) (*Hirsch and Smith, 2006*; *Greulich et al., 2019*). We note that the dynamics of mean cell numbers, *Equations 6 and 7* in the main text, indeed describe an LCS according to this definition. Now we use the following definitions:

- $G(A)$ is the graph of *A*, that is, the graph for which *A* is the adjacency matrix, whose elements $a_{ij}$ give the weight of the links from *i* to *j* ($a_{ij} = 0$ means that no link exists). In the following, we use the terms *graph* and *network* synonymously.
- If in $G(A)$ there exists a path from node *i* to node *j and* from *j* to *i*, then we call those nodes *strongly connected*, $i \equiv j$, which is an equivalence relation. A maximal set of nodes which are are strongly connected with each other are called a *Strongly Connected Component (SCC)* of the graph (the equivalence class of the equivalence relation '$\equiv$').
- The graph $G(A)$ can be decomposed into its $N_S$ SCCs, $S_k$, $k = 1, ..., N_S$ (*Cormen, 2009*), which are sub-graphs associated with an adjacency matrix $A_k$, such that $G(A_k) = S_k$. Since the $A_k$ have non-negative off-diagonal elements, they are Metzler matrices for which the Perron-Frobenius theorem ensures that a unique, simple and real maximal eigenvalue $\mu_k$ exists (*Arrow, 1989*). The eigenvalue $\mu_k$ is called the *dominant eigenvalue* of $S_k$. Associated with this eigenvalue, there is, for all *k*, a positive eigenvector $x^{(k)} = (x_1^{(k)}, x_2^{(k)}, ...)$, that is, one with all entries $x_i^{(k)} > 0$.
- The *condensed graph* of $G(A)$ is the graph where nodes are the SCCs of $G(A)$ and a link from SCC $S_k$ to SCC $S_l$ ($k, l = 1, ..., N_S$) exists if there is is at least one link from a node (in $G(A)$) in $S_k$ to a node in $S_l$.
- If there is a path from SCC $S_k$ to SCC $S_l$, then we call $S_k$ upstream of $S_l$ and accordingly $S_l$ downstream of $S_k$. We note that there can never exist paths from $S_k$ to $S_l$ and from $S_l$ to $S_k$, since otherwise, by definition, their nodes would be strongly connected and both together would form a single SCC (*Cormen, 2009*). Thus, there is a unique hierarchy of SCCs.
- A stationary state $x^*$ of a dynamical system is *Lyapunov stable* if a small initial deviation from $x^*$ leads to a small final deviation $x(t)$ (i.e. $x^*$ is not unstable). More accurately: there exists a constant $C > 0$ such that $|x(t) - x^*| < C|x_0 - x^*|$ for all times *t*, where $x_0 = x(t = t_0)$ is the initial condition, sufficiently close to $x^*$. A stationary state of a linear system that is Lyapunov stable, yet neither asymptotically stable nor has a limit cycle, is *neutrally stable*.
- Homeostasis means that the cell numbers in each state, $n = (n_1, ..., n_m)$, stay on average constant, $\frac{d\bar{n}}{dt} = 0$ (where $\bar{n} = \langle n \rangle$), and that this state is not unstable towards perturbations. This condition corresponds to a Lyapunov-stable stationary state. Note that a linear system, as the one described by *Equations 6 and 7*, main text, cannot have an asymptotically stable state except for the trivial state $\bar{n}^* = 0$, which corresponds to a vanishing cell population. We note that when considering the tissue cell population as a whole, dynamics can be non-linear through interactions between cells and a non-vanishing asymptotically stable state may then exist. However, since single clones do not significantly affect the total configuration of cells in a tissue, the clones compete neutrally, when embedded in a homeostatic cell population, which corresponds to a Lyapunov stable, but not asymptotically stable state. We therefore use Lyapunov stability, a weaker form of stability, to define homeostasis, since an asymptotically stable vanishing state is not a biologically viable state.

Now, for an LCS holds, according to *Greulich et al., 2019*,

### Theorem 1

An LCS, $\dot{x} = Ax$, possesses a non-trivial Lyapunov stable stationary state ($x^* > 0$), if and only if,

1. G(A) does not contain any SCC, $S_k$, with $\mu_k > 0$.
2. There is at least one SCC, $S_k$, with $\mu_k = 0$.
3. There is no path between any two SCCs, $S_k$ and $S_l$, which have $\mu_k = 0$ and $\mu_l = 0$.

Furthermore holds,

## Theorem 2

*All nodes $i$ upstream of an SCC $S_l$ with $\mu_l = 0$ must be empty in the the stationary state, that is, $x_i^* = 0$, if $i$ is upstream of the SCC $S_l$.*

Since *Equation 7*, main text, is an LCS, we can apply theorems 1 and 2 to find conditions for homeostasis, defined by a Lyapunov-stable configuration of mean cell numbers $\bar{n}^* = (\bar{n}_1, \bar{n}_2, ...)$. According to theorem 1 at least one SCC with $\mu_k = 0$ must then exist, and according to theorem 2 the stationary state of nodes upstream of it must be empty, that is, they do not exist in homeostasis. Since the condensed graph of the SCCs does not have cyclic paths, an SCC $S_k$ with $\mu_k = 0$ must therefore always reside at the apex of all non-vanishing cell types. In principle, an acyclic graph may have more than one apex, however, since, by definition, a stem cell clone always starts with a single stem cell, and no other SCC with $\mu = 0$ may be downstream of the latter, we only consider one apex SCC with one initial cell when studying clonal dynamics.

Hence, in the context of homeostatic clonal dynamics, we can assume that there is a single SCC, $S_k$ with $\mu_k = 0$ at the apex of the cell state graph, while all other SCCs, $S_l$ are downstream of it and have $\mu_l < 0$. Since there are no paths from the non-apex SCC to the apex SCC (as the condensed graph is acyclic) we can distinguish the two separate compartments $\mathcal{R}$ (the renewing compartment) consisting of all nodes of the apex SCC, $S_k$, and $\mathcal{C}$ (the committed compartment), consisting of all other nodes, whereby due to $\mu_l < 0$ for all SCCs in $\mathcal{C}$, all progeny of cells in $\mathcal{C}$ will vanish in the long term.

## Stochastic process modelling

### Model description

Since clonal dynamics start, by definition, with a single cell, we use stochastic dynamics to model clones. Thus, we model cell fate dynamics as a continuous-time multi-type branching process (*Haccou et al., 2005*), a Markov process following the rules of *Equations 3-5*, main text. As shown later, without losing generality, here only two types of events are modeled; considering an arbitrary number $m$ of cell states, $X_i$, for $i = 1, ...m$, the model includes

- Cell divisions: a cell in state $X_i$ divides in two cells with rate $\lambda_i$, respectively in state $X_j$ and $X_k$ at a ratio $r_i^{jk}$.

$$X_i \xrightarrow{\lambda_i r_i^{jk}} X_j + X_k, \qquad i,j,k = 1, ..., m, \tag{1}$$

  where $\lambda_i = 0$ if state $X_i$ does not allow division. In this formulation of cell division events, which we use for the generation and numerical simulations of random models, only one division outcome is possible upon division of a particular cell state $X_i$. Nonetheless, multiple division outcomes per state can be implemented as single outcomes if additional *metastates* are introduced, which represent priming of a state $X_i$ towards a certain division outcome option. For example, if in the original model, state $X_i$ has different outcome options, $X_{j_1} + X_{k_1}, X_{j_2} + X_{k_2}, ...$, we can substitute this by, first, transitions from $X_i$ to (new) states $X_{m_1}, X_{m_2}, ...$ and subsequent divisions $X_{m_l} \to X_{j_l} + X_{k_l}$. The use of metastates to model more complex processes is discussed in detail in 'Population Asymmetry model using metastates'.

- Direct state transitions: a cell in state $X_i$ changes to state $X_j$ at a given rate $\omega_{ij}$.

$$X_i \xrightarrow{\omega_{ij}} X_j, \qquad i,j = 1, ..., m; \; i \neq j, \tag{2}$$

  where $\omega_{ij} = 0$ means that no transition from $X_i$ to $X_j$ is possible. Additionally, we include cell loss in this scheme, by treating it as a transition to an additional special state, called hereafter *death* and denoted by $\emptyset$ (cells in this state do not enter in the counting of the total number of cells). In that formulation, the loss rates of the original model are $d_i = \omega_{i\emptyset}$.

These events define a Markov process, which can be represented as a *stochastic network* (***Bang-Jensen and Gutin, 2007***). In this view, each node can be related to a cell state, while the links represent transitions between states via cell divisions and the direct state transitions. It is noted that this stochastic network is different from the network defined in the main text and in 'Conditions for homeostasis' of this SI, which describes the dynamics of *mean* cell number instead. Here, for the stochastic modelling, let us define the adjacency matrix $K$ of this network, through the elements $\kappa_{ij} = \lambda_i 2 r_i^j + \omega_{ij}$ $i,j = 1,...,m$, in which $\kappa_{ij}$ are the total transition rates as defined in the main text. We note that $K$ is related to the matrix $A$ used in the main text by $A = K^T - \Delta$, where $\Delta$ is the diagonal matrix with entries $\delta_i, i = 1,...,m$, as defined in the main text, with the slight difference that here the loss state $\emptyset$ is treated as a separate state. Additionally, it is remarked that in this model interpretation, where only one division option for each state is possible, the term $r_i^j \leq 1$ is not a continuum value, but instead it can only take the values $0, 1/2, 1$ depending on the specific outcome of the division of the cells in state $X_i$. Notably, more than one stochastic network may result in the same matrix $K$, therefore, to uniquely define a process, we distinguish a matrix $D$ which describes cell division events (note that this is possible with just a single matrix as there is only one division option per state) and a matrix $T$ which describes direct transition events. The matrix $K$ is the sum of both, $K = N + T$.

## Generation of random models

To test the behavior of the clonal dynamics in a generic homeostatic model, a large number of random stochastic networks was generated, whereby each stochastic network corresponds to a distinct set of parameters $\lambda_1,...,\lambda_m, \omega_{12},...,\omega_{m\emptyset}$ for the stochastic stem cell fate choice model. The strategy detailed below is based on the following considerations which summarize the key requirements to achieve homeostasis detailed in 'Conditions for homeostasis': (a) each network is composed of Strongly Connected Components (SCCs) that are randomly connected; (b) only one SCC, the one at the apex of the network, forms the renewing compartment, $\mathcal{R}$, (i.e. it is characterized by a dominant eigenvalue $\mu = 0$ with respect to $A$) and all the others form the committed compartment, $\mathcal{C}$, (i.e. they are characterized by a dominant eigenvalues $\mu < 0$). It is further noted that the SCCs of the stochastic network $G(K)$ are the same as those of the matrix $G(A)$, where $A = K^T - \Delta$ defines the dynamics of mean cell numbers. This is, since transposition of an adjacency matrix and altering of diagonal elements does not affect the network topology.

To generate the stochastic network, a two-step process is followed: (1) a large number of (random) SCCs are generated; (2) a condensed network is randomly constructed and filled with randomly picked SCC from step 1.

It is noted that unitary rates are assumed in step (1) and they are successively randomly modified in step (2) to achieve the desired properties of the dominant eigenvalue μ while ensuring randomness.

Focusing now on step (1), that is, the generation of single SCCs, the following procedure is used.

a. The total number of states composing the SCC is defined, indicated as $m_S$. An additional state is added to represent whatever is outside the SCC. In the current analysis, we set $1 \leq m_S \leq 4$.
b. We build separately all the possible combinations of transition and division matrices, indicated hereafter with $M_T$ and $M_D$, respectively. These matrices are ordered for increasing number of transitions $N_T$ and divisions $N_D$. In case GIA networks are generated, the $M_D$ and $M_T$ combinations are filtered, to remain just with those where the division outcome is one cell inside the SCC and one outside the SCC, and where there are only transitions between states within the SCC (i.e. where cell numbers are conserved). From a computational point of view, this process is feasible up to $m_S = 4$.
c. The matrices stored in $M_D$ and $M_T$ are then combined together to form a model (which is completely defined by one matrix in $M_D$ and one in $M_T$); $M_{DT}$ indicates the pool of possible models. This process is done considering separately each $m_S$, $N_T$ and $N_D$. In this step, due to technical limitations given by the high number of possible combinations, if the total number of combinations exceed $5 \cdot 10^4$ then only $10^4$ random matrices from $M_D$ and $M_T$ are combined.
d. Each model in $M_{DT}$ is then processed to check if the corresponding network is a SCC in the first $m_S$ states. If not, then this model is discarded. In case GPA networks are generated, a further check is performed to discard also those models consistent with a GIA network (they

cannot be a priori excluded as done in point 2 for the GIA ones). These pools of models are indicated as $M_{\mathrm{GIA}}$ and $M_{\mathrm{GPA}}$ for the GIA and GPA models, respectively.

e. For each SCC in $M_{\mathrm{GIA}}$ and $M_{\mathrm{GPA}}$, the dominant eigenvalue $\mu$ is estimated. For construction, the generated GIA networks are all characterized by $\mu = 0$, while in general any value can be obtained within $M_{\mathrm{GPA}}$.

f. The SCCs in $M_{\mathrm{GPA}}$ are additionally processed to check whether the network is compatible with homeostasis by tuning the rates. Networks satisfying this condition are additionally stored under a new pool of SCCs, called $M_{\mathrm{GPA}}^*$. If not, then they are discarded when $\mu > 0$ (i.e. for any combination of rates the number of cells in these networks is expected to grow).

This process results in three pools of SCCs classified for $m_S$, $N_T$ and $N_D$ (i.e. number of states, transitions and divisions): (1) $M_{\mathrm{GIA}}$ contains GIA models; (2) $M_{\mathrm{GPA}}^*$ contains GPA models that can be tuned to have $\mu = 0$ and (3) $M_{\mathrm{GPA}}$ contains GPA models characterized by $\mu < 0$ or that can be tuned to meet this condition.

In step (2), the generation of random networks starting from the individual SCCs is implemented as follows.

a. A number of committed SCCs, $N_c$, between 1 and 3 is randomly chosen.

b. $N_c$ SCCs are randomly picked from the pool of models $M_{\mathrm{GPA}}$. The selection is done considering equal probability in $m_S$, $N_T$ and $N_D$. For each SCC, the unitary rates $\alpha$ (where $\alpha$ stands for any rate $\lambda_i$ or $\omega_{ij}$) are modified by multiplying them for random numbers (exponentially distributed with mean $\bar{\alpha} = 1$ and minimum $\alpha_m = 0.3$). Additionally, a threshold on the dominant eigenvalue is set, $\mu_{\max} = -1$; if this condition is not satisfied, then the rates are tuned to meet this requirement while maintaining the rates above the minimum.

c. The committed compartment of the condensed network is generated by randomly connecting all the outgoing components of the $k$-SCC with states in the $l$-SCC for $l = k + 1, .., N_c$. In this way, the transposed adjacency matrix of the stochastic network has triangular block form:

$$
K^T = \begin{bmatrix}
B_1 & & & \\
C_{12} & B_2 & & 0 \\
& & \cdots & \\
C_{1,N_c} & C_{2,N_c} & & B_{N_c} \\
C_{1\emptyset} & C_{2\emptyset} & & C_{N_c,\emptyset} & 0
\end{bmatrix}.
\tag{3}
$$

○ The last SCC is forced to be linked to a single death state.

d. With a similar procedure described in point 2, two SCCs are randomly picked respectively from the pool of SCCs in $M_{\mathrm{GPA}}^*$ and $M_{\mathrm{GIA}}$; the unitary rates are modified (exponentially distributed with mean $\bar{\alpha} = 1$ and minimum $\alpha_m = 0.3$) and, in the GPA case, tuned to meet the condition $\mu = 0$. They represent the renewing part of the network.

e. Two networks (one for the GIA and one for the GPA models) are produced by attaching the selected renewing network upstream the committed one; this is done based on an analogous procedure as described in step 3.

At the end of this process, we have two networks which are different in just the renewing part, being one consistent with the GIA model and the other with the GPA one. In total 2000 networks were built and analyzed.

## Simulation campaign

An extensive simulation campaign was run to model the clone dynamics. The code implemented to numerically simulate the stochastic process defined by events of type 1 and 2 is based on the Gillespie algorithm (*Gillespie, 1977*). Since a clone is by definition the progeny of a single cell, we choose as initial condition a single cell put randomly in a state within $\mathcal{R}$. Concerning the final condition, given the substantial difference in the dynamics in the two models, the final time, indicated by $\tau$, is set equal to 20 times the inverse of the minimum process rate, $\alpha_{\min} = \min(\lambda_1, ..., \lambda_m, \omega_{12}, ..., \omega_{m,\emptyset})$, in the GIA models, and to the time at which the fraction of extinct clones reaches 98% in the GPA models. Note that all critical branching processes, as homeostatic clonal dynamics are, will go extinct almost surely at some point in time (*Haccou et al., 2005*).

To determine the clone size distribution, $10^3$ and $5 \cdot 10^4$ simulations were run respectively in for each GIA and GPA model (in this way, both models result in the same final number of clones when 98% extinction is taken into account).

## Numerical simulation test cases

### Invariant Asymmetry and Population Asymmetry models

To validate the simulation approach, we tested the procedure on simple cell fate models for which analytical results are known, the Invariant Asymmetry (IA) and Population Asymmetry (PA) models. As described in the main text, in the simplest version, these are defined as,

$$S \xrightarrow{\lambda} \begin{cases} S+S & \text{Pr. } r \\ S+D & \text{Pr. } 1\text{-}2r, \\ D+D & \text{Pr. } r \end{cases} \qquad D \xrightarrow{\gamma} \emptyset. \tag{4}$$

In these processes, cells of type $S$ represent the stem cells (called hereafter also progenitor), which divide with stochastic rate $\lambda$, and cells of type $D$ are the differentiated cells, which are shed with rate $\gamma$. While in the PA model the three possible outcomes of the division of a progenitor are controlled by a probability parameter $0 < r \leq 1/2$, in the IA model $r = 0$, meaning that there are strictly asymmetric division and the number of $S$-cells is conserved. It is remarked that in the definition of the stochastic networks given in 'Model description' only one division option for each state is modelled; however, the code implemented for the numerical simulations of the stochastic process allows for an arbitrary number of division options for each state as well (see 'Population Asymmetry model using metastates').

Considering the dynamics at tissue level, the system of ODEs describing the average number of cell $\bar{n}_S$ and $\bar{n}_D$ respectively of type $S$ and $D$ is,

$$\begin{cases} \frac{d\bar{n}_S}{dt} = 0 \\ \frac{d\bar{n}_D}{dt} = \lambda \bar{n}_S - \gamma \bar{n}_D \end{cases}. \tag{5}$$

It is clear that, on average, the number of $S$-cells remains constant. Additionally, in homeostasis, the average total number of $D$-cells stabilizes around a constant value $\bar{n}_D^* = (\lambda/\gamma)\bar{n}_S$ that uniquely depends on the number of stem cells, $\bar{n}_S$ which equals the initial number of stem cells $\bar{n}_{S,0} = \bar{n}_S(t=0)$, Thus, the (Lyapunov stable) stationary state of total cell numbers $\bar{n} = \bar{n}_S + \bar{n}_D$ is given by,

$$\bar{n}^* = \left(1 + \frac{\lambda}{\gamma}\right)\bar{n}_{S,0}. \tag{6}$$

Based on *Equation 6*, the process rates $\lambda$ and $\gamma$ determine the proportion of cells of type $D$ with respect to cells of type $S$. Importantly, there is no difference at tissue level between the IA and PA models.

A distinction is instead evident when we look at the dynamics at the single-cell level, and study the clone size distribution, that is, the distribution of the progeny of a single cell. For the IA model, the number of $S$-cells is strictly constant, and thus the joint probability distribution $P(n_S, n_D)$ of both $S$-cells and $D$-cells, respectively indicated as $n_S$ and $n_D$, is fully determined by the distribution of $D$-cells, $P(n_D)$. The IA model's master equation for $P(n_D)$, considering a single initial cell of type $S$, is given by,

$$\frac{dP(n_D)}{dt} = \lambda P(n_D - 1) + \gamma(n_D + 1)P(n_D + 1) - (\lambda + \gamma n_D)P(n_D). \tag{7}$$

This corresponds to a simple birth-and-death process for which the distribution is Poissonian with mean $\lambda/\gamma$, (*Van Kampen, 1981*).

Considering now the PA model, the master equation is instead given by,

$$\frac{dP(n_S,n_D)}{dt} = \lambda(r(n_S-1)P(n_S-1,n_D) + (1-2r)n_S P(n_S,n_D-1) + r(n_S+1)P(n_S+1,n_D-2))$$
$$+\gamma(n_D+1)P(n_S,n_D+1)$$
$$-(\lambda n_S + \gamma n_D)P(n_S,n_D). \tag{8}$$

In *Antal and Krapivsky, 2010*, an exact result for the distribution of total cell numbers $n = n_S + n_D$ is found when $\lambda = \gamma$ and $r = 1/4$. For different values of the process parameters, the long-term distribution is shown to be Exponential.

Numerical simulations for the clonal dynamics were run, considering the above models and three different sets of test parameters each, indicated as IA# and PA#i for $i = 1, 2, 3$, which are reported in *Appendix 1—table 1*. It is noted that the time unit is arbitrary and therefore omitted. Simulations are based on $10^4$ and $5 \cdot 10^4$ runs respectively for the IA and PA test cases. The initial condition is a single stem cell and the final simulation time, indicated as $\tau$, is equal to 10: this value is well representative of a steady state condition (for the IA test cases) and at which the total extinction of the process is not yet achieved (for PA test cases only). The clone size distribution at $\tau$ in the IA test cases is shown in *Appendix 1—figure 1*: in this figure, each profile is compared to the corresponding Poisson distribution shifted by one (i.e. plus the stem cell). Concerning the results for the PA test cases, they are shown in *Appendix 1—figure 2*. In this case, the profiles are compared to the numerical integration of the master *Equation 8*. Additionally, for the PA# test case, where $\lambda = \gamma$ and $r = 1/4$, the reference analytic solution provided in *Antal and Krapivsky, 2010* is also shown. In general, a good agreement is obtained in all of the cases.

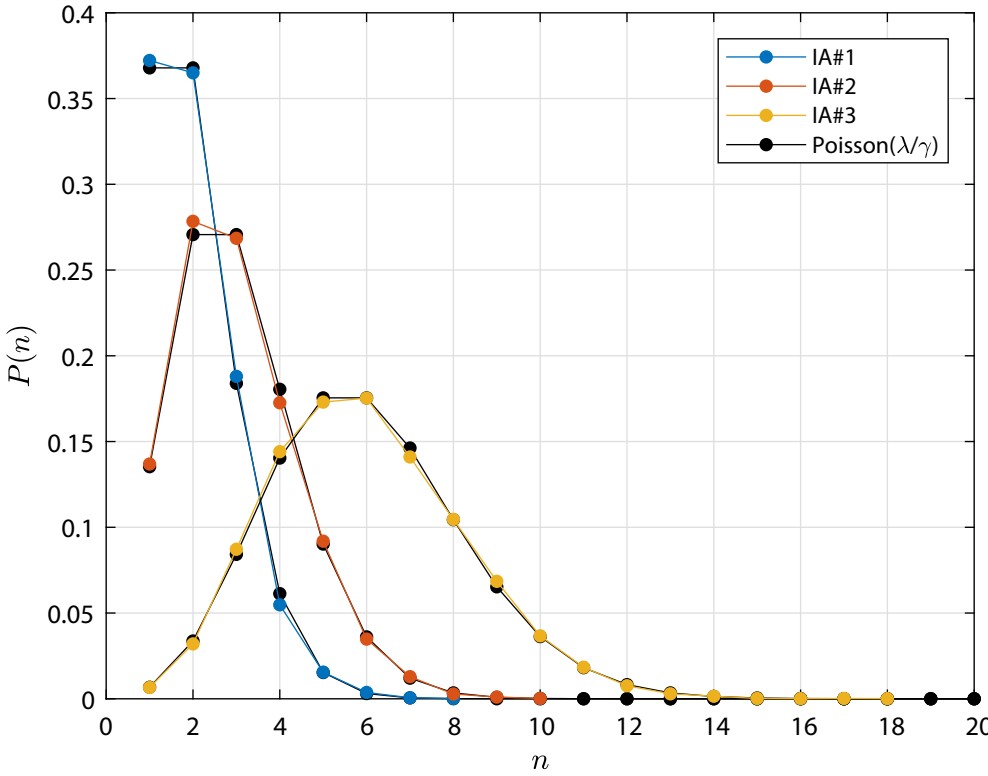

**Appendix 1—figure 1.** Invariant Asymmetry (IA) test cases clone size distribution $P(n)$, that is the distribution of the total number of cells $n$ forming the progeny of a single initial cell in $\mathcal{R}$. For each case, the distribution is shown at $\tau$ (defined in *Figure 2*, main text), which is well representative of the steady state condition. Tested parameters for cases IA#1-3 are provided in *Appendix 1—table 1*; the numerical simulation results are compared to the expected Poisson distribution. The detailed discussion is reported in 'Invariant Asymmetry and Population Asymmetry models'.

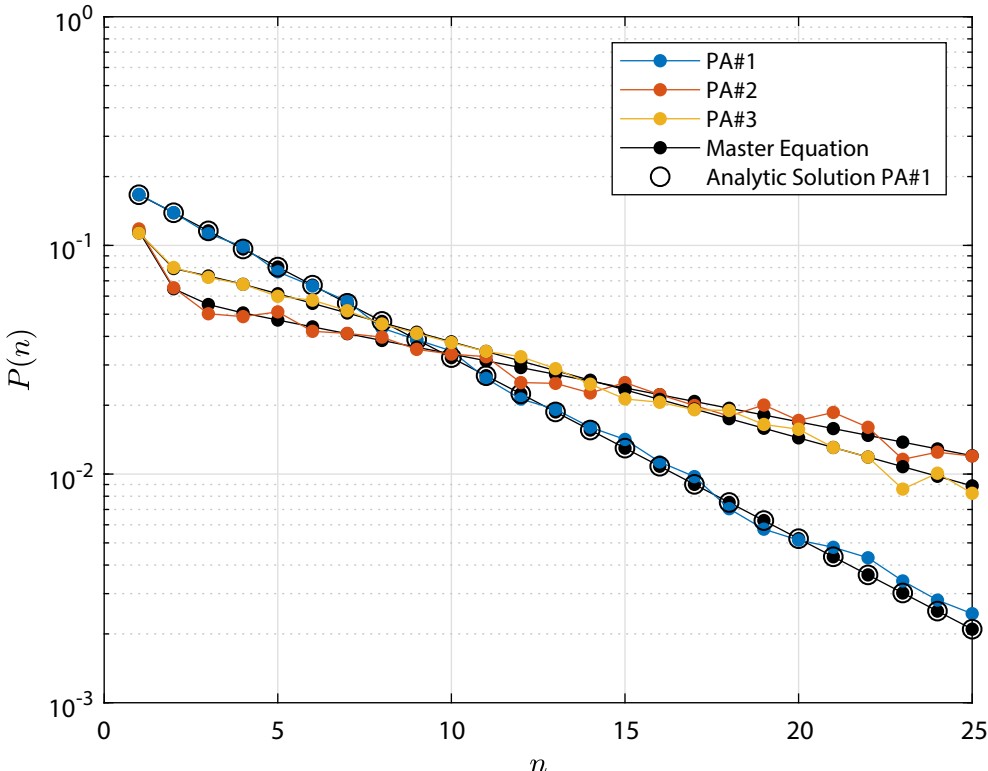

**Appendix 1—figure 2.** Population Asymmetry (PA) test cases clone size distribution $P(n)$, that is the distribution of the total number of cells $n$ forming the progeny of a single initial stem cell. For each case, the distribution is shown at the final time $\tau$, at which the total extinction of the process is not yet achieved. Tested parameters for cases PA#-3 are provided in *Appendix 1—table 1*; the numerical simulation results are compared to the solution of the numerical integration of the master *Equation 8* and, for test case PA#1, also to the reference analytic solution from *Antal and Krapivsky, 2010*. The detailed discussion is reported in 'Invariant Asymmetry and Population Asymmetry models'.

## Population Asymmetry model using metastates

As argued before, we assume in the random model generation that cell division in state $X_i$ has a unique outcome, $X_i \rightarrow X_j + X_k$ (*Equation 1*), since thereby the stochastic process can be uniquely defined by the two matrices $D$ and $T$. To accommodate for the possibility of different division outcomes from the same state $X_i$, as in *Equation 4* and *Equations 3-5* in the main text, we introduce *metastates*, which represent short-lived states that indicate priming for either outcome, from which the cell division outcomes are unique. This is a small modification of the original model, which, however, does not lead to significant deviations if the metastates are traversed sufficiently quickly (which can be assured by a choice of high direct state transition rates in the metastates).

To illustrate this, let us consider the PA model described by 4; instead of having three different outcomes upon division of an S-cell we define the corresponding Metastate (MS) model with three primed states, $M_{1,2,3}$, as,

$$
\begin{aligned}
S &\xrightarrow{\omega_1} M_1, \; M_1 \xrightarrow{\lambda_1} S + S, \\
S &\xrightarrow{\omega_2} M_2, \; M_2 \xrightarrow{\lambda_2} S + D, \\
S &\xrightarrow{\omega_3} M_3, \; M_3 \xrightarrow{\lambda_3} D + D, \\
D &\xrightarrow{\gamma} \emptyset,
\end{aligned}
\tag{9}
$$

in which $S$ and $D$ correspond to the same cell type of the PA model (i.e. the stem and the differentiated cells, respectively), while $M_i$, for $i = 1, 2, 3$, represent the metastates. These states are temporary

states that are used to model each one of the three different possible division options of the *S*-cells. The rates $\lambda_i$ and $\omega_i$, for $i = 1, 2, 3$, are chosen such that the time scales of division and outcome probabilities are the same as in the original PA model:

$$\omega_1/\omega_2 = r/(1-2r), \ \omega_2/\omega_3 = (1-2r)/r, \tag{10}$$

$$\frac{1}{(1/\omega_1 + 1/\lambda_1)} = \lambda r, \ \frac{1}{(1/\omega_2 + 1/\lambda_2)} = \lambda(1-2r), \ \frac{1}{(1/\omega_3 + 1/\lambda_3)} = \lambda r. \tag{11}$$

*Equations 10* assure that outcome probabilities are the same as in the original model, while *Equations 11* are needed to have the same total average time between two consecutive events. As there are six unknowns and only five relations, the following additional equation is added

$$\lambda_1 = \omega_1 \Delta, \tag{12}$$

in which $\Delta$ is an additional parameter that is used to control how fast cells in metastate $M_1$ divide. Low values of $\Delta$ imply that as soon as an *S*-cell transits to the metastate $M_1$, it divides in two *S*-cells. Globally, this results in

$$\begin{aligned}
\omega_1 &= \omega_3 = \lambda r (\Delta + 1)/\Delta \\
\omega_2 &= \lambda(1-2r)(\Delta + 1)/\Delta \\
\lambda_i &= \omega_i \Delta \text{ for } i = 1, 2, 3.
\end{aligned} \tag{13}$$

Numerical simulations for the two models were run and compared, based on the parameters reported in *Appendix 1—table 1*, and specifically the PA#1 and PA#3 test cases. The time unit, which is arbitrary, is omitted. The process rates for the corresponding MS model, which are indicated in the figures as MS#1 and MS#3, are computed based on *Equation 13* and $\Delta = 1/500$. As well as for the PA test cases, the initial condition is one cell of type *S* and the final time, $\tau$, is equal to 10; simulations are based on $5 \cdot 10^4$ trajectories.

**Appendix 1—table 1.** IA and PA test cases simulation parameters (see 'Invariant Asymmetry and Population Asymmetry models').

| Case | $\lambda$ | $\gamma$ | r |
|------|-----------|----------|---|
| IA#1 | 1.0 | 1.0 | - |
| IA#2 | 2.0 | 1.0 | - |
| IA#3 | 5.0 | 1.0 | - |
| PA#1 | 1.0 | 1.0 | 1/4 |
| PA#2 | 2.0 | 1.0 | 1/4 |
| PA#3 | 2.0 | 1.0 | 1/6 |

The mean number of cells in the surviving clones and the extinction probability as function of time (scaled by $\tau$) are shown in *Appendix 1—figure 3*. The clone size distribution at $\tau$ is shown in *Appendix 1—figure 4*. Both MS simulations agree very well with the corresponding PA ones, which justifies the use of metastates for our simulation campaign.

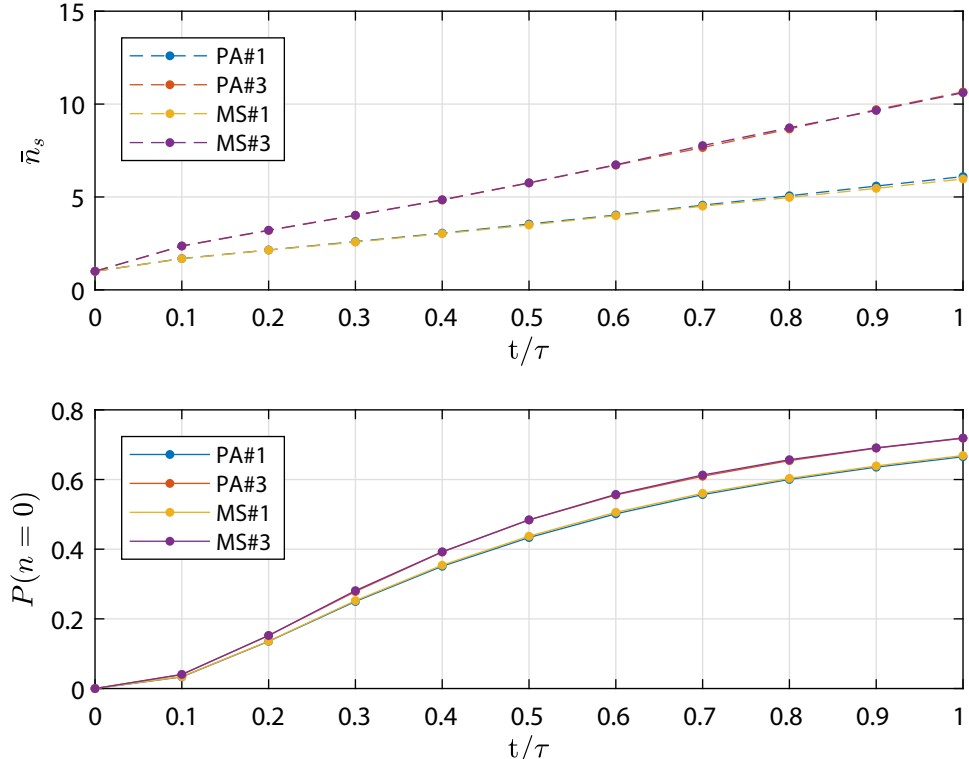

**Appendix 1—figure 3.** Metastate (MS) test cases simulation results in terms of mean number of cells in the surviving clones $\bar{n}_s$ and extinction probability $P(n=0)$ as function of time (scaled by the final simulation time $\tau$). As well as for the PA test cases, at $\tau$ the total extinction of the process is not yet achieved. Profiles from the numerical simulation for cases MS#,3 are compared to the corresponding PA#1,3 test cases which are based on parameters provided in *Appendix 1—table 1*. The detailed discussion is reported in 'Population Asymmetry model using metastates'.

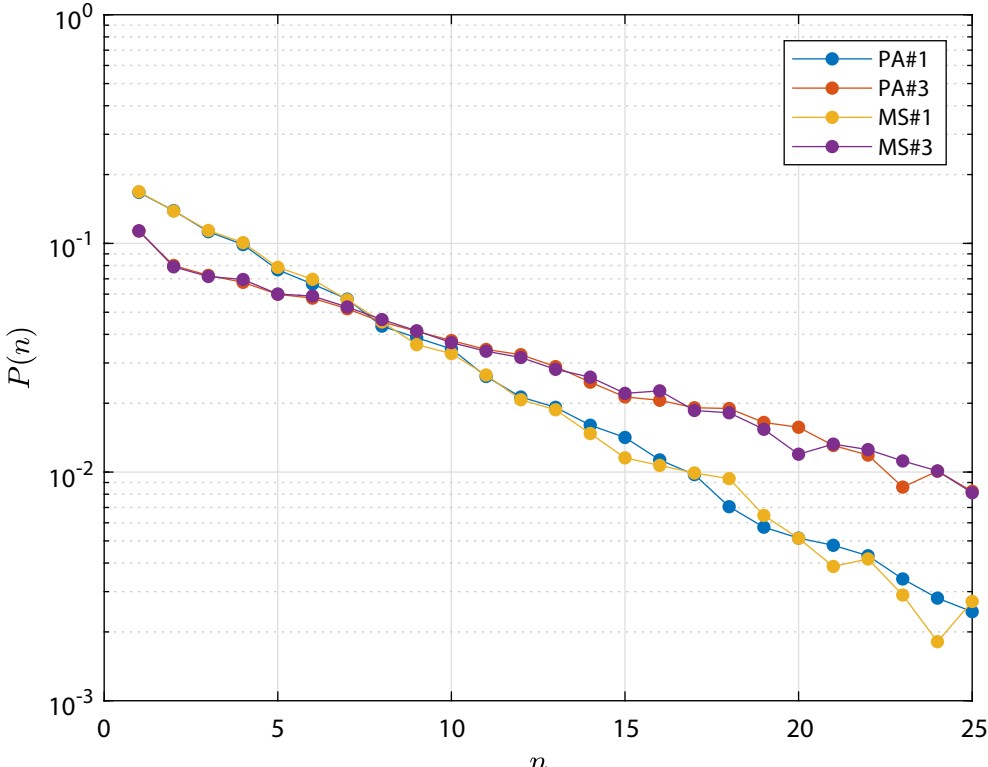

**Appendix 1—figure 4.** Metastate (MS) test cases simulation results in terms clone size distribution $P(n)$, that is the distribution of the total number of cells $n$ forming the progeny of a single initial stem cell. As well as for the PA test cases, the distribution is shown at the final time, $\tau$, at which the total extinction of the process is not yet achieved. Profiles from the numerical simulation for cases MS#,3 are compared to the corresponding PA#1,3 test cases which are based on parameters provided in *Appendix 1—table 1*. The detailed discussion is reported in 'Population Asymmetry model using metastates'.

## Analysis of the Generalized Invariant Asymmetry model

### GIA$^0$ test case: Steady state distribution and limiting behavior

A simple Generalized Invariant Asymmetric model, indicated hereafter as GIA$^0$, was analyzed to identify the causes of the different clone size distribution behaviors observed in the randomly generated models (see main text). Thus, in this section, we study the Markov process defined by,

$$X_1 \xrightarrow{\lambda_1} X_1 + X_2, \ X_2 \xrightarrow{\lambda_2} X_2 + X_2, \ X_2 \xrightarrow{\gamma} \emptyset. \tag{14}$$

Here, the renewing compartment is composed of just a single state $X_1$ and cells in this state asymmetrically divide with rate $\lambda_1$. The committed compartment is formed of state $X_2$; cells in this state can either divide to duplicate, with rate $\lambda_2$, or die, with rate $\gamma$. It is noted that for $\lambda_2 = 0$, this model is reduced to the previously analyzed Invariant Asymmetric (IA) model (see 'Invariant Asymmetry and Population Asymmetry models').

As for the IA model, here the number of cells in state $X_1$, indicated as $n_1$, is conserved. It is therefore sufficient to determine the statistics of $n_2$, defined by the master equation for $P(n_2)$, the probability of having $n_2$ cells in state $X_2$, provided that there are $n_1$ cells in state $X_1$. The master equation is given by,

$$\frac{dP(n_2)}{dt} = \begin{aligned} &-(\lambda_1 n_1 + \lambda_2 n_2 + \gamma n_2)P(n_2) \\ &+(\lambda_1 n_1 + \lambda_2(n_2-1))P(n_2-1) \\ &+\gamma(n_2+1)P(n_2+1), \end{aligned}$$ (15)

also written as,

$$\frac{dP(n_2)}{dt} = \begin{aligned} &-(g(n_2)+r(n_2))P(n_2) \\ &+g(n_2-1)P(n_2-1)+r(n_2+1)P(n_2+1), \end{aligned}$$ (16)

in which $r(n_2) = \gamma n_2$ and $g(n_2) = \lambda_1 n_1 + \lambda_2 n_2$. Considering that we are interested in clonal dynamics, meaning that we start from a single stem cell, $n_1$ is equal to one.

In this simple case, the steady state distribution $P^*(n_2)$, corresponding to the solution of $dP(n_2)/dt = 0$, can be analytically derived. Defining the net flux between states $n_2$ and $n_2 - 1$ as

$$I_{n_2} = r(n_2)P^*(n_2) - g(n_2-1)P^*(n_2-1),$$ (17)

and considering that $I_{n_2+1} = I_{n_2}$ for every $n_2$, it follows that $I_{n_2} = I_0 = r(0)P^*(0) - g(-1)P^*(-1) = 0$, which means that

$$P^*(n_2) = \frac{g(n_2-1)}{r(n_2)}P^*(n_2-1) = \prod_{l=0}^{n_2-1}\frac{g(l)}{r(l+1)}P^*(0),$$ (18)

where $P^*(0)$ is the steady state probability of having 0 cells in state $X_2$. Finally, by applying the conservation of the total probability, $\sum_{n_2=0}^{\infty} P^*(n_2) = 1$, and rearranging the terms we obtain,

$$P^*(n_2) = \left(1-\frac{\lambda_2}{\gamma}\right)^{\lambda_1/\lambda_2}\left(\frac{\lambda_2}{\gamma}\right)^{n_2}\frac{\Gamma\left(\frac{\lambda_1}{\lambda_2}+n_2\right)}{\Gamma(n_2+1)\Gamma\left(\frac{\lambda_1}{\lambda_2}\right)}.$$ (19)

In the main text, we defined the dimensionless parameters $\hat{\lambda}_1 = \lambda_1/\gamma$ and $\hat{\lambda}_2 = \lambda_2/\gamma$, representing the rescaled division rates for cells in state $X_1$ and $X_2$, respectively. For clarity and readability, in this section, we simplify the notation using $p = \hat{\lambda}_1$ and $q = \hat{\lambda}_2$. *Equation 19* is then rewritten as,

$$P^*(n_2) = (1-q)^{p/q}q^{n_2}\frac{\Gamma\left(\frac{p}{q}+n_2\right)}{\Gamma(n_2+1)\Gamma\left(\frac{p}{q}\right)}.$$ (20)

It is noted that while $p$ varies between 0 and $\infty$, $q$ is defined between 0 and 1.

The mean number of cells in each state, indicated respectively as $\bar{n}_1$ and $\bar{n}_2$, satisfies the system of ODEs

$$\begin{cases} \dfrac{d\bar{n}_1}{dt} = 0 \\ \dfrac{d\bar{n}_2}{dt} = \lambda_1 \bar{n}_1 + (\lambda_2 - \gamma)\bar{n}_2 \end{cases}.$$ (21)

Based on this, the steady state average number of cells is

$$\begin{cases} \bar{n}_1^* = 1 \\ \bar{n}_2^* = \dfrac{\lambda_1}{\gamma-\lambda_2} = \dfrac{p}{1-q} \end{cases}.$$ (22)

When the mean number of cells in state $X_2$ is sufficiently large, that is, for large $p$ or in case $q$ is close to one, the discrete distribution given by *Equation 20*, can be approximated by a continuous probability density function $P^*(x_2)$, given by,

$$P^*(x_2) = (1-q)^{p/q} q^{px_2/(1-q)} \frac{\Gamma\left(\frac{p}{q} + \frac{p}{1-q}x_2\right)}{x_2\Gamma\left(\frac{p}{q}\right)\Gamma\left(\frac{p}{1-q}x_2\right)}, \tag{23}$$

in which $x_2 = n_2/\bar{n}_2^*$. We note that *Equation 23* corresponds to *Equation 11* in the main text.

To better understand the distribution for different values of the parameters $p$ and $q$, the limit behavior are analyzed below.

1. **q → 0** (i.e. $\hat{\lambda}_2 \to 0$)

When $q \to 0$, *Equation 20* can be simplified considering that

$$\lim_{q\to 0} \frac{\Gamma\left(\frac{p}{q} + n_2\right)}{\Gamma\left(\frac{p}{q}\right)} \left(\frac{q}{p}\right)^{n_2} = 1, \tag{24}$$

$$\lim_{q\to 0}(1-q)^{p/q} = e^{-p} \tag{25}$$

and

$$\Gamma(n_2 + 1) = n_2!. \tag{26}$$

Thus, the distribution results in

$$\lim_{q\to 0} P^*(n_2) = \frac{p^{n_2}e^{-p}}{n_2!} = \text{Poisson}(p), \tag{27}$$

that is a Poisson distribution with mean equal to *p*. This agrees with what we were expecting considering that when $q = 0$ the model is reduced to the IA model for which the distribution in $n_2$ is known to be poissonian.

Additionally, for large mean number of cells, which are obtained for large *p* (when $q = 0$, then $\bar{n}_2^* = p$), the Poisson distribution tends to a Normal distribution with mean and variance equal to *p*. Therefore,

$$\lim_{(q,p)\to(0,\infty)} P^*(n_2) = \frac{1}{\sqrt{2\pi p}} e^{-\frac{(n_2 - p)^2}{2p}} = \text{Normal}(p, p). \tag{28}$$

Rescaling the distribution, and considering $x_2 = n_2/\bar{n}_2^*$, results in

$$\lim_{(q,p)\to(0,\infty)} P^*(x_2) = \text{Normal}(1, 1/p), \tag{29}$$

that is a Normal distribution with unitary mean and variance equal to $1/p$.

2. **q → 1** (i.e. $\hat{\lambda}_2 \to 1$)

For $q \to 1$ the steady state mean number of cells $\bar{n}_2^* \to \infty$ and *Equation 23* holds. This equation can be rewritten as,

$$P^*(x_2) = q^{p/(1-q)x_2+1} \frac{(1-q)^{p/q}}{q(x_2-1)+1} \frac{\Gamma\left(p\frac{q(x_2-1)+1}{q(1-q)} + 1\right)}{\Gamma\left(\frac{p}{q}\right)\Gamma\left(\frac{p}{1-q}x_2 + 1\right)}. \tag{30}$$

If the Stirling's approximation is applied

$$\Gamma(z+1) = \sqrt{2\pi z}\left(\frac{z}{e}\right)^z, \tag{31}$$

we obtain,

$$P^*(x_2) = \frac{p^{p/q}e^{-p/q}q^{(q-2p)/(2q)}(q(x_2-1)+1)^{p/(1-q)(x_2-1+1/q)-1/2}}{\Gamma\left(\frac{p}{q}\right)x_2^{x_2p/(1-q)+1/2}}. \tag{32}$$

Considering now that

$$\lim_{q\to 1}\frac{(q(x_2-1)+1)^{p/(1-q)(x_2-1+1/q)-1/2}}{x_2^{x_2p/(1-q)+1/2}} = e^{p(1-x_2)}x_2^{p-1}, \tag{33}$$

it follows that

$$\lim_{q\to 1}P^*(x_2) = \frac{p^p}{\Gamma(p)}x_2^{p-1}e^{-px_2} = \mathrm{Gamma}(p,1/p), \tag{34}$$

that is a Gamma distribution with unitary mean and shape parameter given by $p$. Importantly, the Gamma distribution for $p \to \infty$ tends to a Normal distribution with unitary mean and variance $1/p$. For $p = 1$, it corresponds instead to an Exponential distribution with unitary mean.

3. $\mathbf{p} \to \infty$ (i.e. $\hat{\lambda}_1 \to \infty$)

When $p$ is large, the mean number of cells is large for any value of $q$. Thus, *Equation 32* is valid. By applying the Stirling's approximation also to the term $\Gamma(p/q)$, we obtain,

$$P^*(x_2) = \sqrt{\frac{p}{2\pi}}x_2^{-p/(1-q)x_2-1/2}(q(x_2-1)+1)^{p/(1-q)(x_2-1+1/q)-1/2}. \tag{35}$$

This expression can be also rewritten as,

$$P^*(x_2) = \sqrt{\frac{p}{2\pi}}e^{p/(1-q)((x_2-1+1/q)\log(q(x_2-1)+1)-x_2\log(x_2))-1/2(\log(x_2)+\log(q(x_2-1)+1))}. \tag{36}$$

Considering now that $p$ is large, then $-1/2(\log(x_2)+\log(q(x_2-1)+1)) \ll p/(1-q)((x_2-1+1/q)\log(q(x_2-1)+1)-x_2\log(x_2))$, so the term on the right can be neglected. Additionally, for $x_2 \to 1$ the following expansions can be applied:

$$\log(q(x_2-1)+1) = \sum_{k=1}^{\infty}\left((-1)^{k+1}\frac{(q(x_2-1))^k}{k}\right), \tag{37}$$

and

$$\log(x_2) = \sum_{k=1}^{\infty}\left((-1)^{k+1}\frac{(x_2-1)^k}{k}\right). \tag{38}$$

Finally, if we consider that

$$\frac{\left(x_2-1+\frac{1}{q}\right)\sum_{k=1}^{\infty}\left((-1)^{k+1}\frac{(q(x_2-1))^k}{k}\right)-x_2\sum_{k=1}^{\infty}\left((-1)^{k+1}\frac{(x_2-1)^k}{k}\right)}{(x_2-1)^2} = -\frac{1}{2(1-q)}, \tag{39}$$

then *Equation 36* results in

$$\lim_{p\to\infty}P^*(x_2) \simeq \sqrt{\frac{p}{2\pi}}e^{-1/2p(x_2-1)^2} = \mathrm{Normal}(1,1/p), \tag{40}$$

that is a Normal distribution with unitary mean and variance equal to $1/p$.

Importantly, it is noted that the limiting behavior of $P^*(x_2)$ for $q \to 0$ and $q \to 1$ in case of large $p$, are both consistent with the results obtained for $p \to \infty$ and any $q$. In other words, remembering that $p = \hat{\lambda}_1$ and $q = \hat{\lambda}_2$, the steady state distribution for $\hat{\lambda}_1 \to \infty$ and any value of $\hat{\lambda}_2$ is a Normal distribution of unitary mean and variance equal to $1/\hat{\lambda}_1$.

To globally verify these results, numerical simulations of the stochastic process associated with model 14 for different values of $\hat{\lambda}_1$ and $\hat{\lambda}_2$ were run. The following curves were compared:

- Stochastic simulation: distribution at the final simulation time, $\tau$, of the number of cells in state $X_2$. The final time was chosen here as $\tau = 20/\alpha_{\min}$, where $\alpha_{\min} = \min(\lambda_1, \lambda_2, \gamma)$; this value is well representative of a steady state condition. Furthermore, the process rates considered are based on a unitary $\gamma$ (i.e. $\lambda_1 = \hat{\lambda}_1$, $\lambda_2 = \hat{\lambda}_2$ and $\gamma = 1$). It is noted that the time unit is arbitrary and therefore omitted.
- Analytic distribution: based on *Equations 20*, for low mean values, and 23, for large mean values.
- Approximate distributions: Poisson, Gamma and Normal distributions respectively given by *Equations 27, 34 and 40*.

The tested parameters $\hat{\lambda}_1$ and $\hat{\lambda}_2$ are graphically shown in *Appendix 1—figure 5* a contour map showing the expected steady state mean number of cells $\bar{n}_2^*$ over the $(\hat{\lambda}_1, \hat{\lambda}_2)$-parameter plane. The curves from the numerical simulations and the corresponding exact and approximated solutions are shown in *Appendix 1—figure 6*, *Appendix 1—figure 7* and *Appendix 1—figure 8*: the tested conditions are divided into three groups (one figure each) representing the limiting behaviors discussed above. Generally, analytical and numerical results agree very well. This also demonstrates that GIA models can show both peaked and non-peaked distributions, depending on the model parameters.

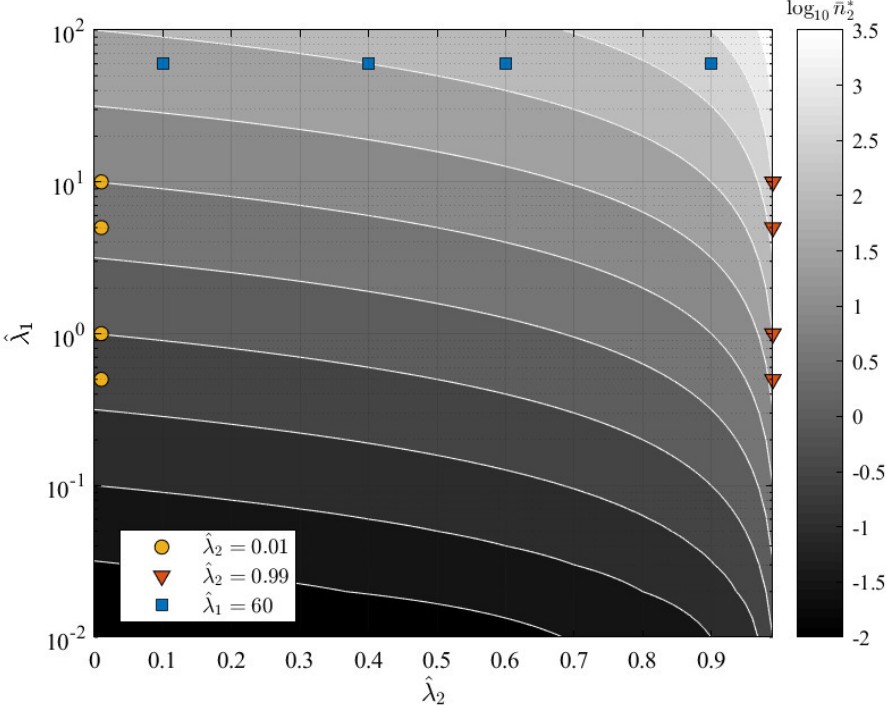

**Appendix 1—figure 5.** GIA$^0$ test case parameters $\hat{\lambda}_1$ and $\hat{\lambda}_2$ over the contour map of the expected steady state mean number of cells in state $X_2$, $\bar{n}_2^*$. The tested conditions are divided in three groups representing the limiting behaviors discussed in in 'GIA$^0$ test case: steady state distribution and limiting behavior', and for which the steady state distribution is shown respectively in *Appendix 1— figure 6*, *Appendix 1—figure 7* and *Appendix 1—figure 8*.

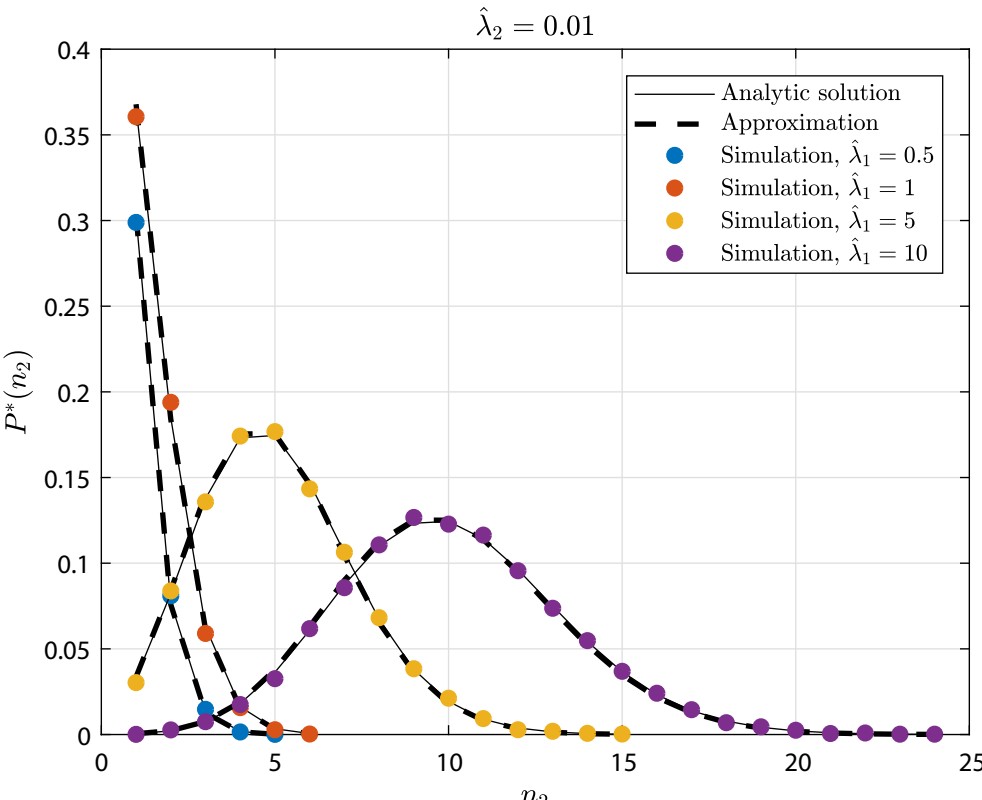

**Appendix 1—figure 6.** $GIA^0$ test case (see $GIA^0$ test case: steady state distribution and limiting behavior') results in terms of steady state distribution $P^*(n_2)$ of the the number of cells in state $X_2$, $n_2$. The tested parameters correspond to the condition $\hat{\lambda}_2 = 0.01$, as representative of the limiting case $\hat{\lambda}_2 \to 0$, and to different values of $\hat{\lambda}_1$. The results from the numerical simulations are compared to the analytic solution (*Equation 20*), and its approximation, that is, the Poisson distribution (*Equation 27*).

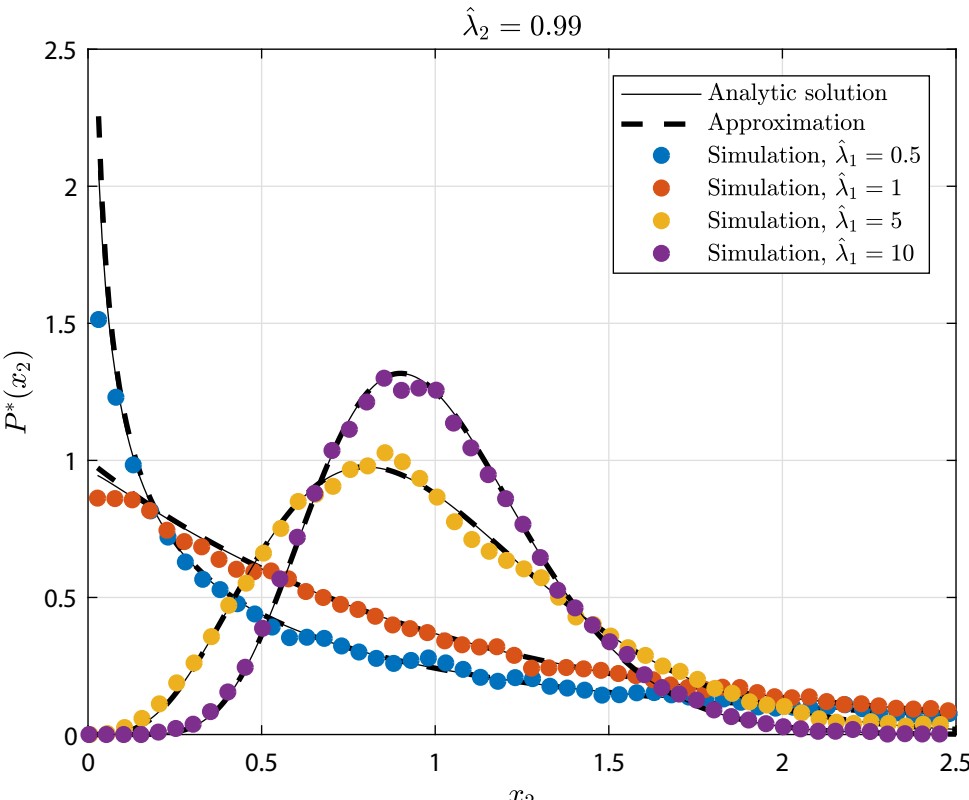

**Appendix 1—figure 7.** $GIA^0$ test case (see 'GIA$^0$ test case: steady state distribution and limiting behavior') results in terms of steady state rescaled distribution $P^*(x_2)$ of the the number of cells in state $X_2$, where $x_2 = n_2/\bar{n}_2^*$. The tested parameters correspond to the condition $\hat{\lambda}_2 = 0.99$, as representative of the limiting case $\hat{\lambda}_2 \to 1$, and to different values of $\hat{\lambda}_1$. The results from the numerical simulations are compared to the analytic solution (*Equation 23*), and its approximation that is the Gamma distribution (*Equation 34*).

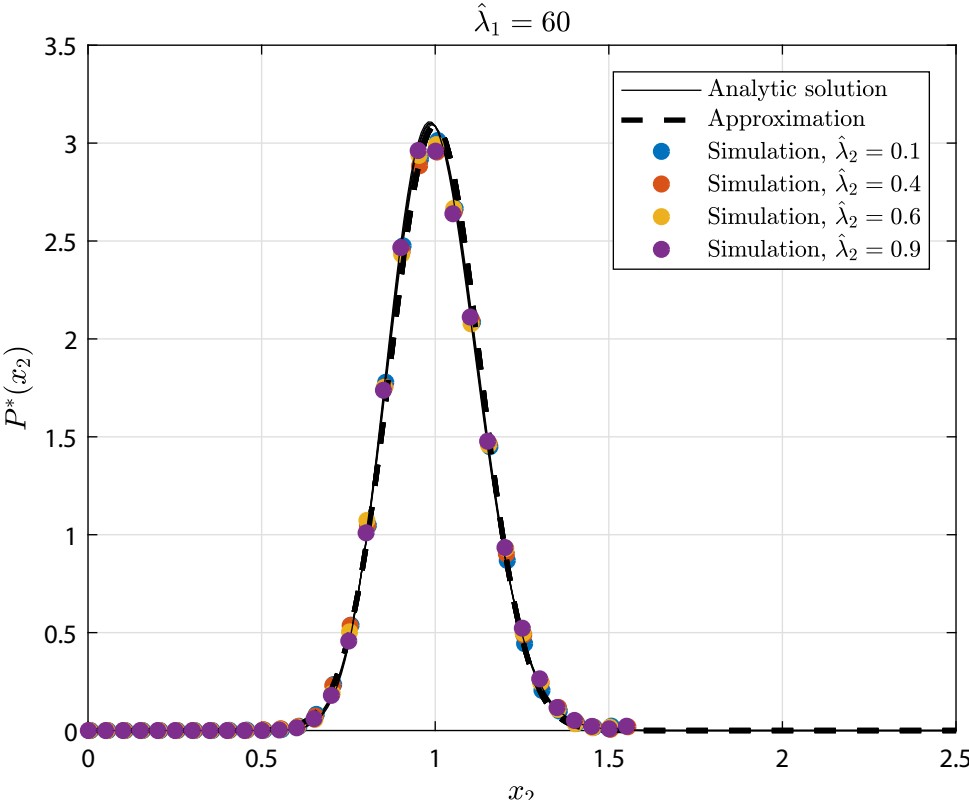

**Appendix 1—figure 8.** $GIA^0$ test case (see 'GIA$^0$ test case: steady state distribution and limiting behavior') results in terms of steady state rescaled distribution $P^*(x_2)$ of the the number of cells in state $X_2$, where $x_2 = n_2/\bar{n}_2^*$. The tested parameters correspond to the condition $\hat{\lambda}_1 = 60$, as representative of the limiting case $\hat{\lambda}_1 \to \infty$, and to different values of $\hat{\lambda}_2$. The results from the numerical simulations are compared to the analytic solution (**Equation 23**), and its approximation that is the Normal distribution (**Equation 40**).

## Approximation of generic GIA models

As shown in the main text, a generic GIA model can be expressed in terms of the compartments $\mathcal{R}$ and $\mathcal{C}$ (**Equation 9** in the main text). We note that the the GIA$^0$ model discussed in the previous section corresponds to the general compartment dynamics of GIA models, **Equation 9**, main text, if the dynamics of compartments are assumed to be Markovian. Thus, we can treat the GIA$^0$ model as a Markovian approximation of generic GIA models. In this section, we test this approximation numerically.

To this end, we first wish to relate the effective (non-Markovian) rates $\lambda_{R,C}$ and $\gamma_C$ of a generic GIA model to the rates of the Markovian approximation, the GIA$^0$ model. We refer to this model – the GIA$^0$ model matched to the effective rates of a particular more complex GIA model – as the *equivalent model* to the latter. The equivalent rates $\lambda_R$, $\lambda_C$ and $\gamma_C$ are computed considering the same steady state condition in terms of mean number of cells. To this aim, we rewrite the dynamics of mean cell numbers, **Equation 7** in the main text, in block form as,

$$\begin{cases} \dfrac{d\bar{\mathbf{n}}_R}{dt} = A_{RR}\bar{\mathbf{n}}_R \\ \dfrac{d\bar{\mathbf{n}}_C}{dt} = A_{CR}\bar{\mathbf{n}}_R + A_{CC}\bar{\mathbf{n}}_C \\ \dfrac{d\bar{n}_{\emptyset}}{dt} = A_{\emptyset C}\bar{\mathbf{n}}_C \end{cases} , \tag{41}$$

in which $\bar{n}_{R,C}$ denote the vectors of mean cell numbers of states restricted to compartments $\mathcal{R},\mathcal{C}$, respectively, and $n_{\emptyset}$ the number of lost cells (not considered for total cell numbers and homeostasis

condition). It is noted that $A_{RC} = \mathbf{0}$, since there cannot be links from $\mathcal{C}$ to $\mathcal{R}$. Also $A_{\emptyset R} = \mathbf{0}$ as we do not consider loss from $\mathcal{R}$ (see main text for the arguments).

Thus, summing up all the components in each compartment, $\bar{n}_R = \sum_i (\bar{\mathbf{n}}_R)_i = 1$ and $\bar{n}_C = \sum_i (\bar{\mathbf{n}}_C)_i$, results in

$$
\begin{cases}
\dfrac{d\bar{n}_R}{dt} = 0 \\[2mm]
\dfrac{d\bar{n}_C}{dt} = \sum_i (A_{CR}\bar{\mathbf{n}}_R)_i + \sum_i (A_{CC}\bar{\mathbf{n}}_C)_i \\[2mm]
\dfrac{d\bar{n}_\emptyset}{dt} = A_{\emptyset C}\bar{\mathbf{n}}_C
\end{cases} . \tag{42}
$$

The equivalent parameters are then estimated from the steady state condition $\bar{\mathbf{n}}_X^*$ and $\bar{n}_X^*$, for $X = R, C, \emptyset$, as,

$$
\lambda_R = \sum_i (A_{CR}\bar{\mathbf{n}}_R^*)_i, \; \gamma_C = \frac{\sum_i (A_{\emptyset C}\bar{\mathbf{n}}_C^*)_i}{\bar{n}_C^*} \text{ and } \lambda_C = \gamma_C - \frac{\lambda_R}{\bar{n}_C^*}. \tag{43}
$$

The applicability of this approximation was evaluated by comparing the clone size distribution obtained from the random GIA models (generated as described in 'Generation of random models' and analyzed in the main text) with that from the corresponding equivalent GIA$^0$ model with parameters $\hat{\lambda}_1 = \hat{\lambda}_R = \lambda_R/\gamma_C$ and $\hat{\lambda}_2 = \hat{\lambda}_C = \lambda_C/\gamma_C$. The values of $\hat{\lambda}_1$ and $\hat{\lambda}_2$ for all the GIA random models are shown in **Appendix 1—figure 9** in the contour map of the expected mean number of cells in $\mathcal{C}$ (in compartment $\mathcal{R}$ there is always one single cell). In general, $\hat{\lambda}_1$ remains below five and $\hat{\lambda}_2$ is spread between zero and one. As measure of the error of the equivalent model, $\epsilon$, we choose the maximum difference between the distributions of a particular random GIA model and that of the corresponding equivalent model, relative to the peak of the distribution of the random model. For low mean cell numbers, the distribution is compared to **Equation 20**; for large mean number instead, the rescaled distribution is compared to **Equation 23**. A threshold on the mean cell number equal to 10 was chosen to distinguish between the two cases. This relative error $\epsilon$ as function of $\hat{\lambda}_2$ is presented in **Appendix 1—figure 10**, where it is evident that large errors are obtained only for large values of this parameters. Some illustrative cases, representative of different value of $\hat{\lambda}_2$, were selected and their distribution is shown in **Appendix 1—figure 11**, **Appendix 1—figure 12** and **Appendix 1—figure 13**. The following considerations are made:

- Two cases for $\hat{\lambda}_2 < 0.2$ are presented in **Appendix 1—figure 11**. In these cases, the distribution obtained from the random models agrees with the analytic solution from the equivalent model, which in turn is well approximated by a Poisson distribution. As expected, larger deviations between the equivalent model's analytic solution and the approximation are noted for increasing values of $\hat{\lambda}_2$. In general, all the models in this range are well approximated by the equivalent model.

- The two cases presented in **Appendix 1—figure 12** have $\hat{\lambda}_2 > 0.8$, for which the Gamma distribution is an approximation of the equivalent model's analytic solution. The distribution in some cases (see for instance the top figure), presents some deviations with respect to the equivalent model. However, globally a good agreement is obtained in most of the cases (failing ratio, based on a 0.5 maximum error is 21.7%).

- Two cases in an intermediate range $0.2 < \hat{\lambda}_2 < 0.8$ are shown in **Appendix 1—figure 13**. Again, the equivalent model's analytic solution is well representative of the distribution (failing ratio, based on a 0.5 maximum error is 3.2%). It is noted that for such values of $\hat{\lambda}_2$ an approximation of the equivalent model analytic solution is not available.

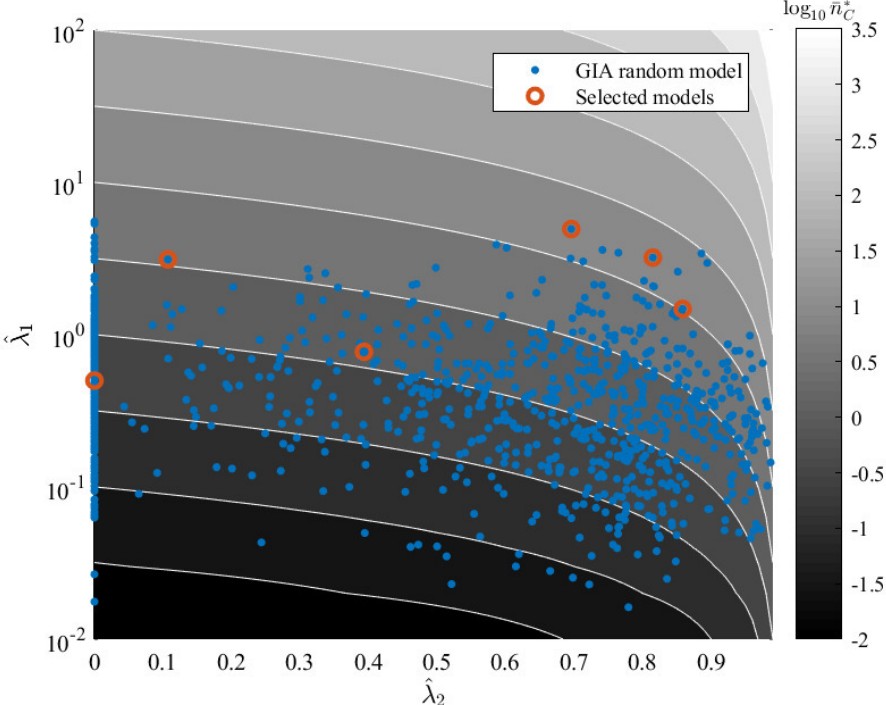

**Appendix 1—figure 9.** GIA random models (generated as described in 'Generation of random models' and analyzed in the main text) equivalent parameters $\hat{\lambda}_1 = \hat{\lambda}_R$ and $\hat{\lambda}_2 = \hat{\lambda}_C$ (see section Approximation of generic GIA models) over the contour map of the expected steady state mean number of cells in the committed compartment, $\bar{n}_C^*$. Some illustrative cases, for which the steady state distribution is shown in *Appendix 1—figure 11*, *Appendix 1—figure 12* and *Appendix 1—figure 13*, are highlighted.

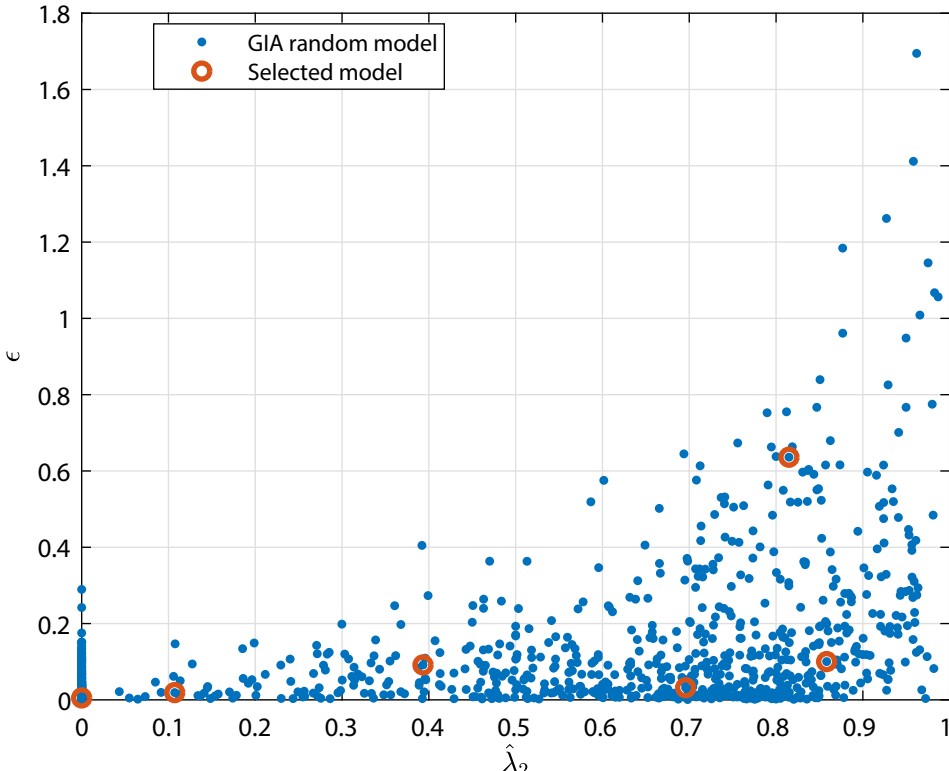

**Appendix 1—figure 10.** Relative error of the the equivalent model approximation, $\epsilon$, (see definition in 'Approximation of generic GIA models') as function of $\hat{\lambda}_2 = \hat{\lambda}_C$ for the GIA random models (generated as described in 'Generation of random models' and analyzed in the main text). The selected cases correspond to some illustrative cases for which the steady state distribution is shown in *Appendix 1—figure 11*, *Appendix 1—figure 12* and *Appendix 1—figure 13*.

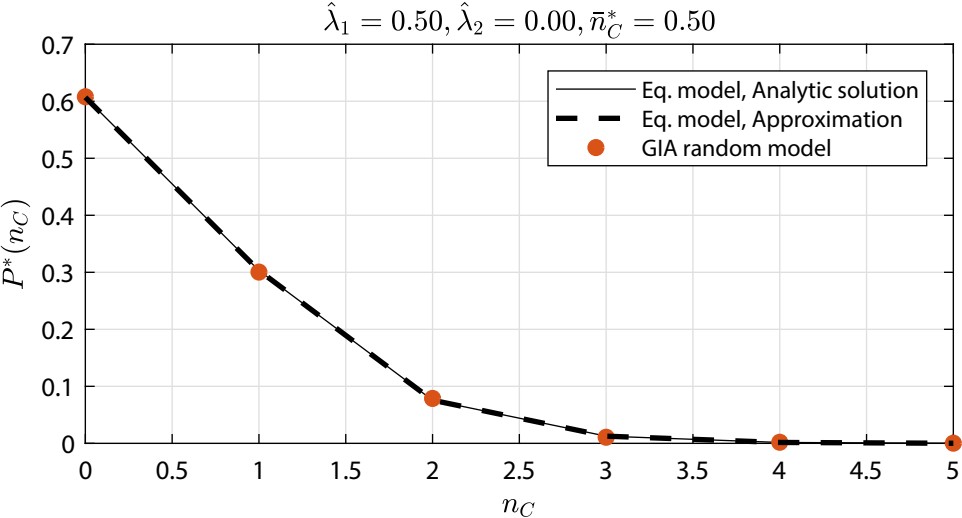

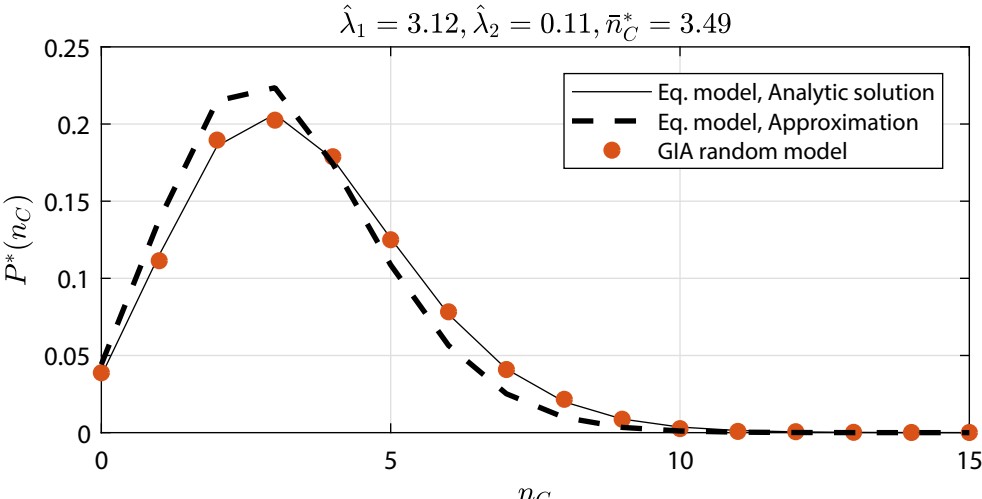

**Appendix 1—figure 11.** GIA random models selected cases (see *Appendix 1—figure 9* and *Appendix 1—figure 10*) where $\hat{\lambda}_2 < 0.2$: the steady state distribution $P^*(n_C)$ of the number of cells in the committed compartment, $n_C$, is compared to that of the equivalent model (Equation model in the legend) analytic solution and its approximation for low $\hat{\lambda}_2$ (i.e. the Poisson distribution, Poisson $(\hat{\lambda}_1)$). Results discussion is reported in 'Approximation of generic GIA models'.

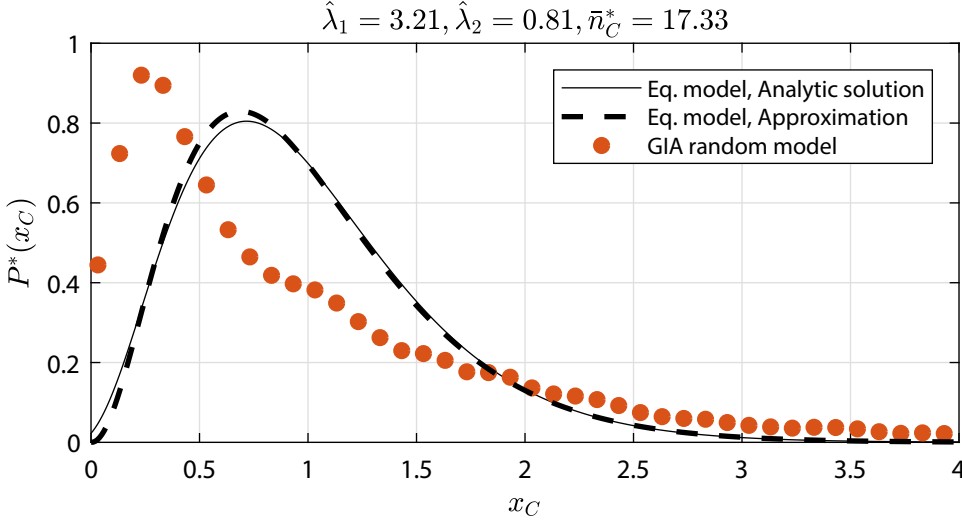

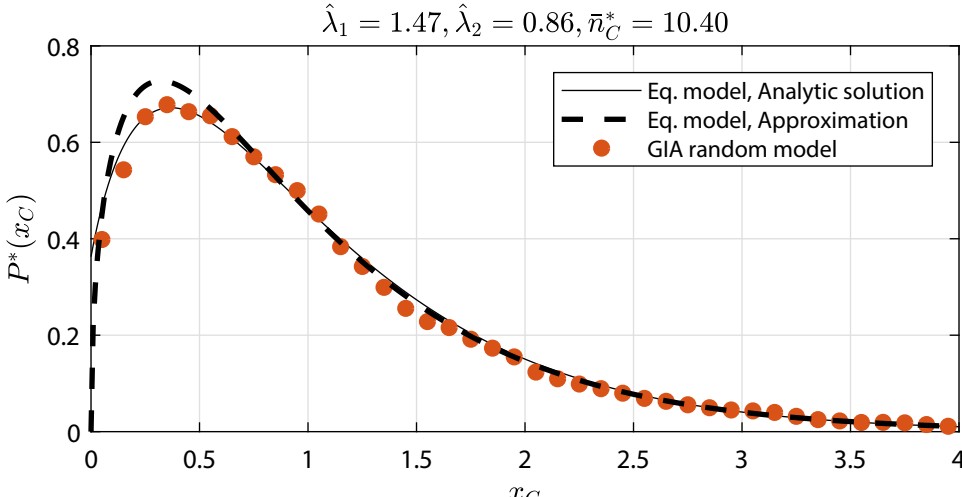

**Appendix 1—figure 12.** GIA random models selected cases (see *Appendix 1—figure 9* and *Appendix 1—figure 10*) where $\hat{\lambda}_2 > 0.8$: the steady state rescaled distribution $P^*(x_C)$ of the number of cells in the committed compartment, where $x_C = n_C/\bar{n}_C^*$, is compared to that of the equivalent model (Equation model in the legend) analytic solution and its approximation for high $\hat{\lambda}_2$ (i.e. the Gamma distribution, Gamma $(\hat{\lambda}_1, 1/\hat{\lambda}_1)$). Results discussion is reported in 'Approximation of generic GIA models'.

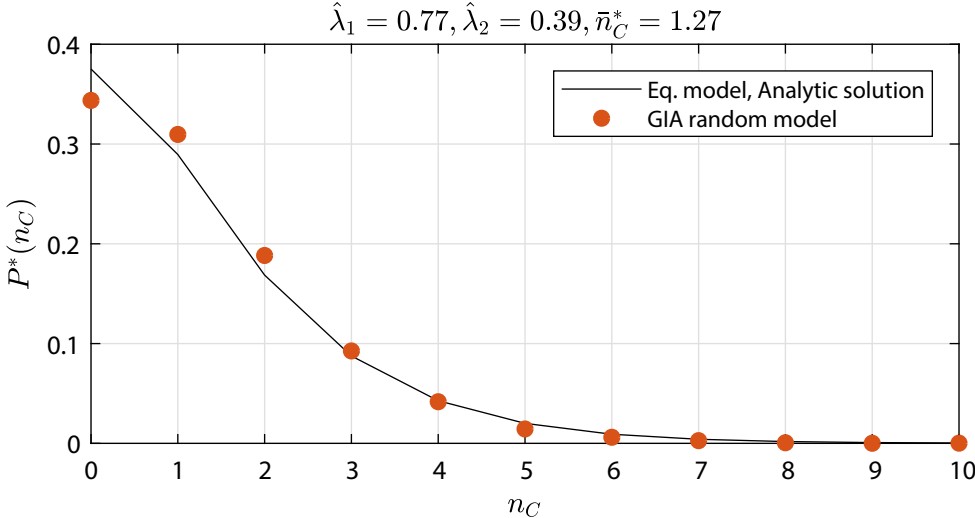

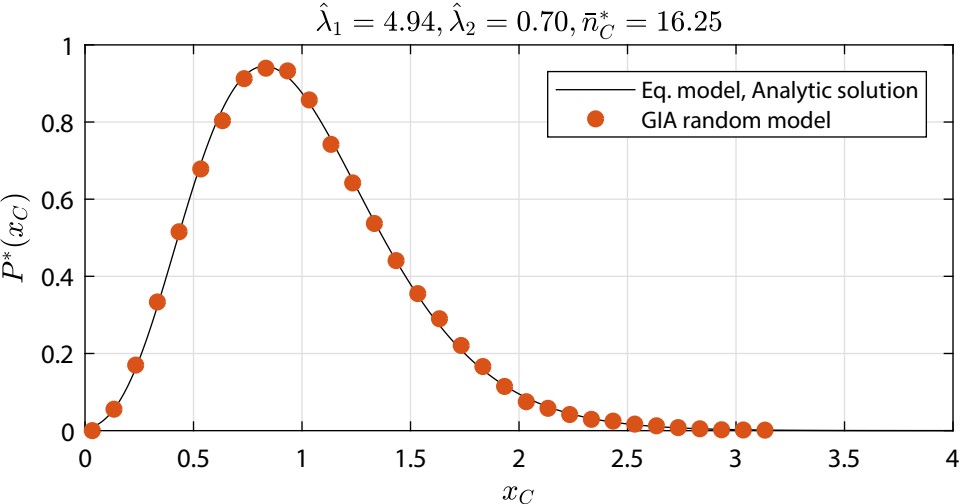

**Appendix 1—figure 13.** GIA random models selected cases (see *Appendix 1—figure 9* and *Appendix 1—figure 10*) where $0.2<\hat{\lambda}_2<0.8$: the steady state distribution $P^*(n_C)$ (or the rescaled distribution $P^*(x_C)$) of the number of cells in the committed compartment, $n_C$ (or in the rescaled case $x_C = n_C/\bar{n}_C^*$), is compared to that of the equivalent model (Equation model in the legend) analytic solution. Results discussion is reported in 'Approximation of generic GIA models'.

Thus, in most of the tested cases the equivalent model is able to catch the behavior of a generic random GIA model, and thus represents a good approximation (global failing ratio, based on a 0.5 maximum error is 6%). In the cases where the equivalent model does not yield a good approximation, the internal structure of the $\mathcal{R}$ and $\mathcal{C}$ compartments become relevant and subsequent events that affect $n_R$ and $n_C$ become dependent on each other, and thus are non-Markovian.

## GIA model for large $\hat{\lambda}_R$

To test the behavior of a generic GIA model in case of large $\hat{\lambda}_R$, the GIA random models (generated as described in 'Generation of random models' and analyzed in the main text) were modified by changing the process rates associated to the renewing compartment to achieve $\hat{\lambda}_R = 30$. To this aim, considering that infinite solutions are possible, we applied a global search method, and more specifically a Genetic Algorithm (*Goldberg, 1989*). We therefore setup an optimization problem, where the process parameters are the optimization variables and the cost function is the error of the current $\hat{\lambda}_R$ with respect to the target.

The envelope of curves obtained in all the random models and some illustrative profiles are shown in *Appendix 1—figure 14*. A reference Normal distribution, characterized by unitary mean and variance equal to $1/\hat{\lambda}_R = 1/30$ is also reported: this curve corresponds to the distribution expected in the equivalent model for which $\hat{\lambda}_1 = \hat{\lambda}_R$. Deviations become relevant, when the internal structure of compartments in a random model leads to subsequent events that are not independent from each other. These effects alter the variance of the Normal distribution. In fact, *Figure 4* in the main text is based on the same simulation results, but in this case the rescaling is done considering both the mean number of cells and its variance (a Normal distribution is a two-parameter distribution).

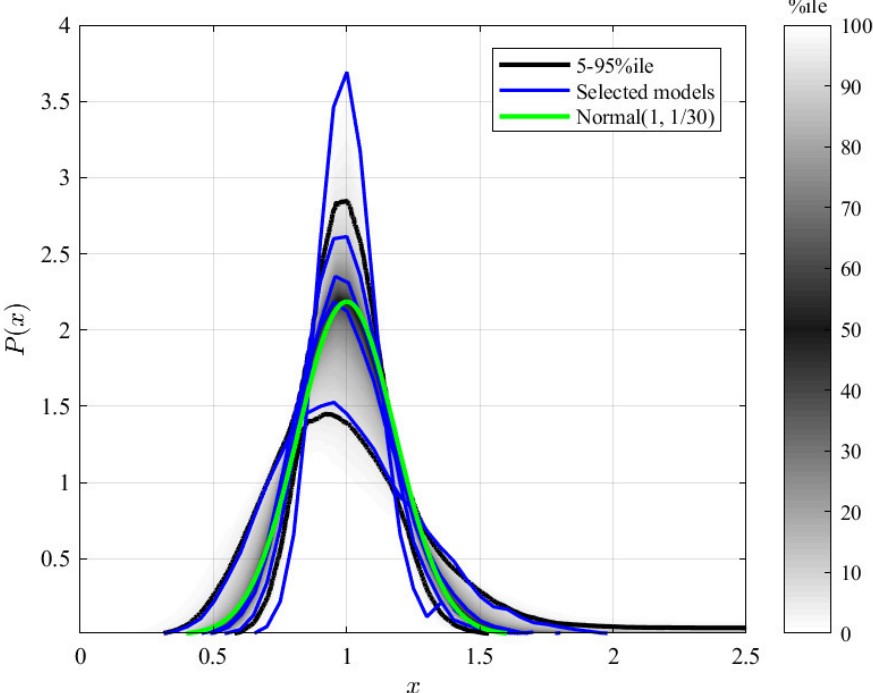

**Appendix 1—figure 14.** Rescaled clone size distribution for the random GIA models when $\hat{\lambda}_R = 30$ at the final simulation time, which corresponds to $20/\alpha_{\min}$ ($\alpha_{\min}$ is the minimum process rate). The grey shade represents the percentile of all the simulations (black lines limit the 5-95%ile range); the blue curves correspond to some illustrative selected simulations. A reference curve corresponding to a Normal distribution of unitary mean and variance equal to $1/\hat{\lambda}_R = 1/30$ is shown in green. Distributions of the total number of cells $n$ are scaled by the mean number of cells $\bar{n}$, being $x = n/\bar{n}$. Simulations for which the final condition (20 times the inverse of the minimum process rate) is not achieved (due to computational limitations) are not included, resulting in 922 models. Results discussion in provided in 'GIA model for large'.

## GIA$^B$ test case: bimodal distribution

In the previous subsection we increased $\lambda_R$ in a way which assures that other parameters within $\mathcal{R}$ stay of the same order of magnitude. Here, we address the question what happens if some parameters within $\mathcal{R}$ are much smaller than parameters of $\mathcal{C}$, such as $\gamma_C$. For that purpose, we study another simple GIA model, let us call it GIA$^B$, as a modification of the GIA$^0$ test model defined by 14. In the GIA$^B$ model the renewing compartment is composed by two states $X_1$ and $X_2$, instead of only one. Cells in these states divide asymmetrically (i.e. one daughter cell remains within the renewing compartment while the other enters the committed compartment) or change state between $X_1$ and $X_2$ (*cell state switching*) while still remaining within the renewing compartment. The committed compartment of the system is composed just by a single state, $X_3$, and cells in this state either duplicate or die (as the previous state $X_2$ in *Equation 14*). This corresponds to the model

$$X_1 \xrightarrow{\lambda_1} X_1 + X_3, \ X_2 \xrightarrow{\lambda_2} X_2 + X_3, \ X_1 \xrightarrow{\omega_{12}} X_2, \ X_2 \xrightarrow{\omega_{21}} X_1, \ X_3 \xrightarrow{\lambda_3} X_3 + X_3, \ X_3 \xrightarrow{\gamma} \emptyset. \tag{44}$$

In this model, the effective parameters as defined in 'Approximation of generic GIA models', $\lambda_R = \lambda_1 P_1^* + \lambda_2 P_2^*$, where $P_i^* = \frac{\omega_{ji}}{\omega_{ij} + \omega_{ji}}$, $i,j = 1,2, i \neq j$, and $\gamma_C = \gamma$. As before, we define the non-dimensionalized parameters $\hat{\lambda}_R = \lambda_R/\gamma_C$ and here we also define $\hat{\omega} = \omega_{12}/\gamma_C$, and further the parameter ratios $a = \lambda_1/\lambda_2$ and $b = \omega_{12}/\omega_{21}$. In the following, we test this model for different values of $a$ and $\hat{\omega}$ as reported in *Appendix 1—table 2*, while fixing $\hat{\lambda}_R = 30$, which is our main scaling parameter, as well as $\hat{\lambda}_C = 0$ and $b = 1$.

**Appendix 1—table 2.** GIA$^B$ test case simulation parameters (see 'GIA$^B$ test case: bimodal distribution').

| Case | $\hat{\omega}$ | $\lambda_1/\lambda_2$ |
|---|---|---|
| GIA$^B$#1 | $3 \ 10^1$ | 1 |
| GIA$^B$#2 | $3 \ 10^{-2}$ | 1 |
| GIA$^B$#3 | $3 \ 10^2$ | 10 |
| GIA$^B$#4 | $3 \ 10^1$ | 10 |
| GIA$^B$#5 | $3 \ 10^0$ | 10 |
| GIA$^B$#6 | $3 \ 10^{-1}$ | 10 |
| GIA$^B$#7 | $3 \ 10^{-2}$ | 10 |

The rescaled distribution of the number of cells in the committed compartment $\mathcal{C}$ (i.e. in state $X_3$), $n_C$, obtained at the final simulation time $\tau$, is shown in *Appendix 1—figure 15*. A value of $\tau$ equal to $20/\alpha_{\min}$ (where $\alpha_{\min}$ is the minimum of all rate parameters) was chosen to assure that the steady state is reached. Considering first the test cases GIA$_B$#1 and GIA$_B$#2 according to *Appendix 1—table 2*, which are characterized by $a = 1$ (i.e. there is no difference in the division timescales for the two renewing states), they both lead to a Normal distribution, independently on the value assumed by $\hat{\omega}$. Test cases GIA$_B$#3 to GIA$_B$#7 instead are all characterized by $a = 10$, and different orders of magnitude for $\hat{\omega}$ are tested. The distribution in these cases is Normal until $\hat{\omega} \geq \hat{\lambda}_R/10$ (see cases GIA$_B$#3 to GIA$_B$#5); when $\hat{\omega}$ is significantly lower than $\hat{\lambda}_R$, then bimodality emerges (see cases GIA$_B$#6 and GIA$_B$#7). Looking at the extreme case, GIA$_B$#7, cells in each renewing state, if analyzed independently, would result in a Poisson distribution in the committed compartment with different mean values (low for the slow-dividing state and large for the fast-dividing one). Thus, globally the distribution is in line with a bimodal distribution computed as

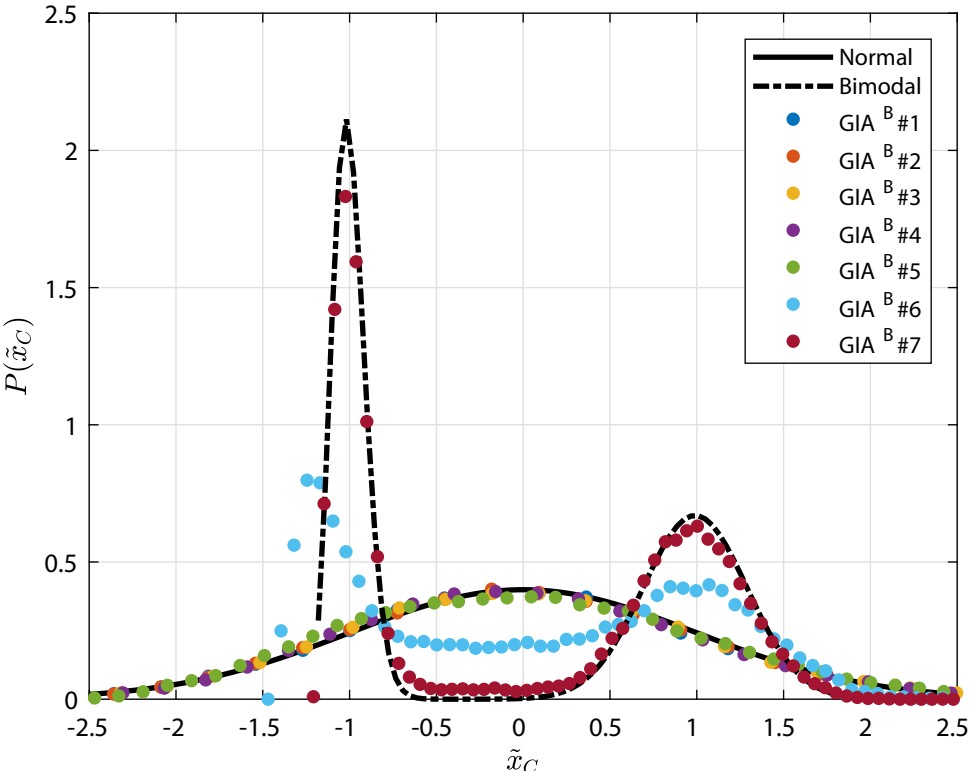

**Appendix 1—figure 15.** Rescaled distribution of the cells number in the committed compartment in the $GIA^B$ test cases at time $\tau$, which is $20/\alpha_{\min}$ ($\alpha_{\min}$ is the minimum process rate). The distributions $P(\tilde{x}_C)$ of the number of cells in the committed compartment $n_C$ is rescaled considering that $\tilde{x}_c = (n_C - \bar{n}_C)/\sigma_{n_c}$, where $\sigma_{n_c}$ is the variance of $n_c$. In addition to the stochastic simulation results for different settings (see *Appendix 1—table 2*), the reference Normal and bimodal distributions are also shown. Results discussion is provided in 'GIA$^B$ test case: bimodal distribution'.

$$P(n) = \beta \mathrm{Poisson}(\hat{\lambda}_R^{(1)}) + (1-\beta)\mathrm{Poisson}(\hat{\lambda}_R^{(2)}), \tag{45}$$

in which $\beta$ is the mixing parameter, computed as,

$$\beta = \frac{\bar{n} - \bar{n}_2}{\bar{n}_1 - \bar{n}_2}, \tag{46}$$

and the parameters $\hat{\lambda}_R^{(i)}$ and $\bar{n}_i$ for $i = 1, 2$ correspond to the parameter $\hat{\lambda}_R$ and to the mean number of cells of a system in which the renewing compartment would be composed just by state $X_i$. The total mean number of cells is instead indicated by $\bar{n}$. The bimodal distribution given by *Equation 45* is indicated as a black dashed-dotted line in *Appendix 1—figure 15*.

## Analysis of the Generalized Population Asymmetry model

In the main text, it is shown that GPA models predict asymptotically, for large times $t$, the same rescaled clone size distribution, that is, an Exponential distribution of unitary mean.

In *Appendix 1—figure 16*, the 50%tile distribution of all the GPA models analyzed is shown at different levels of extinction (which are related to the different time points), showing a gradual convergence to the expected Exponential distribution.

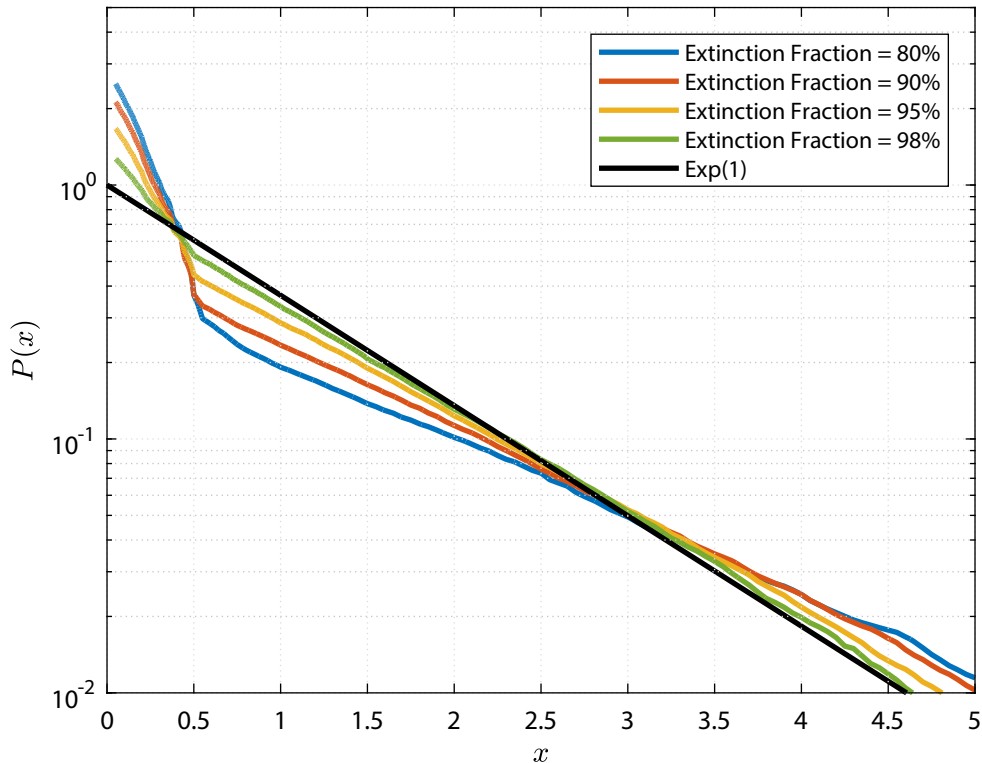

**Appendix 1—figure 16.** Clonal size distribution (corresponding to the 50%ile curve) in the GPA random models at different extinction fraction (i.e. different time). The curves are compared to the expected Exponential distribution (see 'Analysis of the generalized Population Asymmetry model').

Thus, the Markov approximation to all GPA models, *Equation 12* in the main text (the *equivalent model* of GPA models), becomes accurate for sufficiently large *t* and no significant deviations are observed. This also means that for large *t*, the distribution is independent of the choice of parameters, since only the mean value of surviving clones, $\bar{n}_s$, depends on parameters, which however, does not affect the rescaled distribution in terms of $x = \frac{n}{\bar{n}_s}$. We can therefore abstain from an extended study of different parameter regimes. This is in contrast to the GIA model class where distributions depend sensitively in the choice of parameters if we are not in the scaling regime of large $\hat{\lambda}_R$, and the non-Markovian nature of GIA models can become relevant, as we showed in the previous section.

## Asymptotic clone size distributions: Mathematical analysis

In the previous two sections, we studied numerically how a Markovian representation can approximate general cell fate models (GIA and GPA) models. Here, we study from an analytical view point how generic GIA and GPA models converge to the respective limiting distributions, for large time *t* (GPA models) and large $\hat{\lambda}_R$ (GIA models).

Similar to 'Approximation of generic GIA models', we define $n_R$ and $n_C$ as the cell number vectors (here: actual cell numbers of the stochastic model, not mean cell numbers) restricted to the states of compartments $\mathcal{R}$ and $\mathcal{C}$, respectively. We further define the accumulated cell numbers $n_R = \sum_i (n_R)_i$ and $n_C = \sum_i (n_C)_i$ in $\mathcal{R}$ and $\mathcal{C}$, respectively. Considering $n_R$ and $n_C$ as *observables* of our compartment model, this corresponds to a *Hidden Markov Model* in that the dynamics of the observables are not Markovian, yet they are entirely determined by a set of states which follow a Markov process.

### General dynamics of *C*-cells for GIA and GPA models

### Comments on the effective rate parameter $\lambda_R$

For general GIA and GPA models in the compartment representation of *Equation 9*, main text, the effective rate parameter $\lambda_R$ (i.e. the frequency of cell divisions in $\mathcal{R}$ per cell), is defined similar as in

'Approximation of generic GIA Models', yet, here we take into account that $\lambda_R$ can depend on time via the – not necessarily stationary – distribution of cells within $\mathcal{R}$ (since the process is non-Markovian). Hence, in these more general terms, we define $\lambda_R(t) = \sum_{i \in \mathcal{R}} \lambda_i P_i^R(t)$ where $P_i^R(t) = \frac{\bar{n}_i(t)}{\bar{n}_R(t)}$ is the probability of a single cell to be in state $i$ at time $t$. $P_i^R(t)$ may variate after each event $E$, as the conditional probability $P^R|_E$, provided that an event $E$ has just occurred, differs from the stationary state distribution.

In homeostasis, where the number of $R$-cells must on average stay constant, $\lambda_R$ is also the rate, per $R$-cell, at which $C$-cells are created from $R$-cells, via events $R \to R + C, R \to C + C$, or direct transition, $R \to C$. Thus, the total rate of $C$-cells being created from the $R$-cells by such events – let us call them $RC$-events – is $\lambda_R n_R$. While the non-Markovian nature of the process does not assure that such events are distributed exponentially, we can state that, by definition, the number of such creation events in a time period $\Delta t$, $N_{RC}$, has mean value $\langle N_{RC}(\Delta t) \rangle = \int_0^{\Delta t} \lambda_R(t) n_R(t)\, dt$.

While, $\lambda_R(t)$ may in principle depend on time, we note that when internal rates of $\mathcal{R}$ are fast compared to the time period $\Delta t$ above (an *internal rate* of $\mathcal{R}$ is a rate $\omega_{ij}$ where states $i,j$ are both in $\mathcal{R}$), then $\lambda_R(t)$ fluctuates quickly and we can make an adiabatic approximation, replacing $\lambda_R(t)$ by its average $\bar{\lambda}_R = \sum_{i \in \mathcal{R}} \lambda_i P_i^R$, where $P_i^{R*} = \frac{\bar{n}_i^*}{\bar{n}_R^*}$ is the steady state value of $P_i^R(t)$ (this corresponds for GIA models to the definition of $\lambda_R$ in 'Approximation of generic GIA models'). This is fulfilled in our simulations of large $\hat{\lambda}_R$, since internal rates, such as $\hat{\omega}$ defined in 'GIA$^B$ test case: bimodal distribution', scale with $\hat{\lambda}_R$ when $\lambda_R \to \infty$ (see 'GIA model for large'). Hence, the time scales of internal rates are substantially smaller than the relevant time scale $\Delta t = 1/\bar{\gamma}_C$, the lifetime of generated $C$-cells. Therefore, when comparing with simulation results, it is generally appropriate to assume that $\lambda_R(t) \approx \bar{\lambda}_R$ is constant. In the following subsection, we will discuss this case. The case when internal rates are slower than the time scale $\gamma_C$ is discussed in the subsequent subsection.

## Asymptotic distributions of C-cells

Each $C$-cell created by an $RC$-event initiates a sub-clone within $\mathcal{C}$, defined through its progeny, which then follows the dynamics of $\mathcal{C}$. Such sub-clones evolve independently of each other (a defining characteristic of branching processes [*Haccou et al., 2005*]). Let us call the number of cells of a sub-clone created by an $RC$-event at time $t_i$, which evolves over time $t$, as $\xi_i(t)$. We denote two $RC$-events which happen at the same time via a symmetric division of type $R \to C + C$ by different indices $i$ and $i+1$, yet with $t_i = t_{i+1}$. Therefore, the total number of cells in $\mathcal{C}$ is the sum of independent random numbers $\xi_i$,

$$n_C(t) = \sum_{i=1}^{N_{RC}} \xi_i(t) \tag{47}$$

Note that the random numbers $\xi_i(t)$ are not identically distributed, since their statistics depend on the time point of the $i$-th $RC$-event. In particular, the mean value, $\bar{\xi}_i(t - t_i) = \langle \xi_i(t) \rangle$ and variance $\sigma_\xi^2(t - t_i) = \langle (\xi_i(t) - \bar{\xi}_i)^2 \rangle$ depend on the time passed since the respective $RC$-event at time $t_i$. Thus, we cannot apply the central limit theorem in its original form to the sum of random numbers, *Equation 47*. However, a variation of the central limit theorem states that sums of non-identically distributed random variables, $\sum_i \xi_i$, converge to normally distributed random variables, if mean and variance of $\xi_i$ are finite, *and* they fulfill *Lindeberg's condition* (*Billingsley, 1995*).

The (strict) Lindeberg's condition is said to be fulfilled for a sequence of random numbers $\xi_i, i = 1, ..., N$, if

$$\max_i \frac{\sigma_i^2}{\sigma_N^2} \to 0, \text{ for } N \to \infty \tag{48}$$

where $\sigma_i^2 = \langle (\xi_i - \bar{\xi}_i)^2 \rangle$ and $\sigma_N^2 = \sum_{i=1}^{N} \sigma_i^2$. If this is fulfilled, then $n_C = \sum_{i=1}^{N} \xi_i$ converges for $N \to \infty$ to a random variable that is normal distributed.

To show that the $\xi_i$ fulfill Lindeberg's condition, we note that $\xi_i(t - t_i)$ follow a sub-critical multi-type branching process, for which $\bar{\xi}_i(t) \to 0$ for $t \to \infty$, which is assured since the eigenvalues of the adjacency matrix of $\mathcal{C}$ are all negative (since dominant eigenvalues of all SCCs in $\mathcal{C}$ are negative

[astrom_murray_feedback_book]). For multi-type branching processes, the variance $\sigma^2$ is proportional to the mean value, hence $\sigma_i^2(t - t_i) \sim \bar{\xi}(t - t_i)$. Therefore, $\sigma_i^2 \to 0$ for $t \to \infty$, hence it is bounded, i.e there exists $C > 0$ such that $\sigma_i^2(t) < C$ for all $t$. Furthermore, since initially, at $t = t_i$, $\bar{\xi}_i(t_i) = 1$, we know that there exist $t_1 > 0$ and $\delta > 0$ such that $\bar{\xi}_i(t) > \delta$ for $t - t_i < t_1$. Now we recall that, since here we assume the validity of the adiabatic approximation discussed in the previous subsection, the number of $RC$-events within a time period $\Delta t$ is $N_{RC}(\Delta t) \sim \lambda_R \int_0^{\Delta t} n_R(t') \, dt'$. For generic $\lambda_R$, $N_{RC}$ is finite and thus is $\sigma_N$, since all $\sigma_i(t) \to 0$ for large $t$. However, for $\lambda_R \to \infty$ or $n_R \to \infty$, we get that $N_{RC}(t_1) \sim \bar{\lambda}_R n_R \to \infty$ and thus $\sigma_N^2 = \sum_{i=1}^{N_{RC}} \sigma_i^2(t) > N_{RC} \delta \to \infty$. On the other hand, all $\sigma_i^2 < C$, which means that all $\frac{\sigma_i^2}{\sigma_N^2} < \frac{C}{\sigma_N^2} \to 0$ for $\lambda_R \to \infty$ or $n_R \to \infty$. Hence, Lindeberg's condition is fulfilled if $\lambda_R \to \infty$ or $n_R \to \infty$ and thus, $n_C$ becomes normally distributed,

$$n_C(t) = \sum_i^{N_{RC}} \xi_i(t) \to \text{Normal}(\text{mean} = \bar{n}_C, \text{variance} \sim \bar{n}_C) \tag{49}$$

The variance scales with $n_C$ since variances of independent random numbers add linearly and each $\sigma_i^2 \sim \bar{\xi}_i$. The exact value of $\bar{n}_C$ and the pre-factor of the variance of $n_C$ in this limit depend on the (non-Markovian) model details.

### Deviations from a normal distribution in the asymptotic case

The arguments leading to *Equation 49* hold for large $\hat{\lambda}_R$ if the internal rates of $\mathcal{R}$ are comparable to $\bar{\lambda}_R = \sum_i \lambda_i \frac{\bar{n}_i^*}{\bar{n}_R^*}$, which is satisfied for all cases we sampled randomly for numerical simulations, see 'GIA model for large'. However, if internal rates of $\mathcal{R}$ are much smaller than $\lambda_R$, then the adiabatic approximation $P_i^R(t) \approx \frac{\bar{n}_i^*}{\bar{n}_R^*}$ does not apply and $\lambda_R(t)$ may vary slower than the time scale $1/\bar{\gamma}_C$. For example, consider a GIA model in which $\mathcal{R}$ can be decomposed into two sub-compartments, say $\mathcal{R}_1$ and $\mathcal{R}_2$, whereby any rates $\omega_{ij}, \omega_{ji}$ with $i \in \mathcal{R}_1, j \in \mathcal{R}_2$ have $\omega_{ij}, \omega_{ji} \ll \lambda_R$, as the example discussed in 'GIA$^B$ test case: bimodal distribution'. Then, the single cell in $\mathcal{R}$ (note that always $n_R = 1$ in GIA models) may spend long time periods in $\mathcal{R}_1$ and $\mathcal{R}_2$ respectively. Now, if $\bar{\lambda}_{R_1} = \sum_{i \in \mathcal{R}_1} \lambda_i \frac{\bar{n}_i}{\bar{n}_{R_1}} \neq \sum_{i \in \mathcal{R}_2} \lambda_i \frac{\bar{n}_i}{\bar{n}_{R_2}} = \bar{\lambda}_{R_2}$, then, for time periods exceeding $1/\bar{\gamma}_C$, the effective asymmetric division rates are $\bar{\lambda}_{R_1}$ and $\bar{\lambda}_{R_2}$ respectively, and during these time periods the distribution of $n_C$ cells has mean $\bar{n}_C^{(1)} \sim \bar{\lambda}_{R_1}$ and $\bar{n}_C^{(2)} \sim \bar{\lambda}_{R_2}$ respectively. Hence, the total clone size distribution will be the mix of two Normal distributions with mean $\bar{n}_C^{(1)}$ and $\bar{n}_C^{(2)}$, respectively, that is, a bimodal distribution. This scenario is discussed in 'GIA$^B$ test case: bimodal distribution', for the specific case of two states in $\mathcal{R}$.

### GIA models

In GIA models, the number of $R$-cells is conserved, and in particular, for clones, we have $n_R = 1$ for all times. Hence, the rate of $RC$-events is simply $\lambda_R$. Now, if internal rates are fast and $\lambda_R \to \infty$, then $n_C$ becomes normally distributed, as argued above. Hence, also $n = n_R + n_C = 1 + n_C$ follows a Normal distribution, with mean $n_C + 1$ instead.

Nonetheless, if internal rates are less than $\gamma_C$ then bimodal distributions may be observed, as discussed in 'GIA$^B$ test case: bimodal distribution'.

### GPA models

The dynamics of GPA models read, in compartment formulation,

$$R \xrightarrow{\lambda_R} \begin{cases} R + R & \text{Pr.} \, r_{RR} \\ R + C & \text{Pr.} \, 1 - r_{RR} - r_{CC}, \\ C + C & \text{Pr.} \, r_{CC} \end{cases} \tag{50}$$

$$R \xrightarrow{\omega_{RC}} C, \quad C \xrightarrow{\lambda_C} C + C, \quad C \xrightarrow{\gamma_C} \emptyset \tag{51}$$



Since the dynamics of $R$-cells do not depend on $C$-cells, we can first consider the formers' dynamics separately. In homeostasis, where $\lambda_R r_{RR} = \lambda_R r_{CC} + \omega_{RC}$, we have thus for $R$-cells,

$$n_R \xrightarrow{\lambda_R r_{RR} n_R} n_R \pm 1 \tag{52}$$

This is a simple continuous time branching process with two offspring; yet it is non-Markovian: subsequent events may be correlated, since each event imbalances the internal distribution $P_i^R$ of cells in the compartment $\mathcal{R}$. Yet, as for $C$-cells, we can write the number of $R$-cells as a sum of independent (but not identically distributed) random variables. Let us consider for each $R$-cells, born at time $t_i$, the random variable $\xi_i^R$ describing its 'survival' state, that is, $\xi_i^R = 1$ if that cell is still in $\mathcal{R}$, and $\xi_i^R = 0$ if that cell has left $\mathcal{R}$ via symmetric differentiation, $R \to C + C$ or direct transition, $R \to C$. Essentially, the random numbers $\xi_i^R$ are the 'branches' of the branching process. Since these events do not depend on other cells, the random numbers $\xi_i^R$ are independent of each other, and thus,

$$n_R(t) = \sum_{i=1}^{N_b(t)} \xi_i^R(t) \ , \tag{53}$$

is a sum of independent, not identically distributed random variables. Here, $N_b(t)$ is the total number of birth events occurring at rate $\lambda_R r_{RR} n_R$, $R \to R + R$, up to time $t$. Since $\xi_i^R(t) \leq 1$ and $\xi_i^R(t = t_i) = 1$, we can argue analogue to above for **Equation 49** that the sequence of $\xi_i^R$ fulfills Lindeberg's condition and thus $n_R$ converges to a Normal distribution, whereby the mean value $\bar{n}_R = 1$ (since due to homeostasis the mean number is constant and the initial condition is $n_R(t = 0) = 1$). Hence, the probability to have $n_R$ cells in $\mathcal{R}$ is

$$P(n_R) \propto e^{-\frac{(n_R - 1)^2}{2\sigma_R^2}} \sim e^{-\frac{n_R^2}{2\sigma_R^2}} \text{ for } n_R \gg 1 \ . \tag{54}$$

However, here, the variance $\sigma_R^2$ is a random variable itself: Since the $\xi_i^R$ are independent, $\sigma_R^2 = \sum_{i=1}^{N_b(t)} \sigma_i^2$, where $\sigma_i^2 = \langle (\xi_i^R - \bar{\xi}^R)^2 \rangle$, and where $N_b(t)$ is a random variable. The random numbers $\xi_i^R$ can only have the values $\xi_i = 1$ or $\xi_i^R = 0$ and they follow a simple death process, so for $\xi^R = 0$, it must be $\sigma_i^2 = 0$, while for $\xi_i^R = 1$, the variance must be finite, let's say, $\sigma_i^2 = \beta(t) > 0$ where $\beta$ can in principle depend on time, yet is not known (it depends on the non-Markovian details of the model). Hence,

$$\sigma_R^2 = \sum_{i=1}^{N_b(t)} \beta(t) \xi_i^R = \beta(t) n_R \tag{55}$$

since the number of summands with $\xi_i^R = 1$ is the number of surviving $R$-cells, that is, $n_R$. Substituting $\sigma_R^2 = \beta(t) n_R$ into **Equation 54** gives,

$$P(n_R) \sim e^{-\frac{n_R^2}{2\beta(t) n_R}} = e^{-\frac{n_R}{2\beta(t)}} \tag{56}$$

This is an Exponential distribution with mean value $\bar{n}_R = \langle n_R \rangle = 2\beta(t)$. Finally, when we enforce normalisation of the probability distribution, we get,

$$P(n_R) = \frac{1}{\bar{n}_R(t)} e^{-\frac{n_R}{\bar{n}_R(t)}} \text{ for } n_R \gg 1 \ . \tag{57}$$

Eventually, we also have to 'add' the $C$-cells. Since for $t \gg 1$, also $n_R \gg 1$, individual events $n_R \to n_R \pm 1$ do not significantly affect the distribution of $R$-cells, $P_i^R = \frac{\bar{n}_i}{\bar{n}_R}$ (in contrast to the case of $n_R = 1$ for GIA models), and hence we can assume the adiabatic approximation discussed above, where $P_i^R \approx P_i^{R*}$ and thus $\lambda_R \approx const$. Therefore, $C$-cells are distributed according to a Normal distribution with mean $\bar{n}_C$ and variance $\sigma_{n_2}^2 \sim \bar{n}_C \sim \lambda_R n_R$. As argued in the main text, the mean value of $n_R$, conditionend on survival of a clone, $n_R > 0$, must grow over time, without bound if $t \to \infty$. Therefore, we can generally assume that $n_R \gg 1$, and hence both $\bar{n}_C \sim n_R \to \infty$ and $\sigma_C^2 \sim n_R \to \infty$. However, if we express the

clone size in form of a rescaled variable $x = \frac{n}{\bar{n}_s}$ ($\bar{n}_s$ is the mean of surviving clones) we can write $x = x_R + x_C$ with $x_R = \frac{n_R}{\bar{n}_s}$ and $x_C = \frac{n_C}{\bar{n}_s}$, and note that the rescaled standard width of the distribution of $x_C$, $\sigma_{x_C} = \frac{\sigma_C}{\bar{n}} \sim \frac{\sqrt{\bar{n}_C}}{\bar{n}_R + \bar{n}_C} \sim \frac{\sqrt{\bar{n}_R}}{n_R}$ vanishes for $t \to \infty$. Therefore, $x_C$ is effectively a constant in that limit, $x_C \approx \bar{x}_C \propto x_R$. Hence, also $x = x_R + x_C \propto x_R$ and thus, the rescaled clone size, $x = \frac{n}{\bar{n}_s}$, is distributed according to an Exponential distribution (here: a probability density function) with unit mean, and after renormalisation, we get that

$$P(x) = e^{-x} \text{ for } t \to \infty. \tag{58}$$

This distribution is indeed observed in all our simulations of GPA models for large $t$.

