## [Decision Letter]

**Acceptance summary:**

This work builds on several recent high-profile applications of statistical physics to infer the mode of stem cell fate choice in mammalian tissues based on clone size distributions obtained via cell lineage tracing. The work makes a solid contribution especially in clarifying how a broad class of models conceptually and quantitatively fits with the population asymmetry or the invariant asymmetry model. Such universality argument may have been already taken for granted by physicists/mathematicians but may have not been obvious to biologists who have specific systems in mind that have much richer cell state structure than the original simplified models.

Through a detailed mathematical analysis, the authors demonstrate that this large set of models can, in homeostatic tissues, be decomposed into two 'universality classes' in terms of their predicted asymptotic clone size statistics. This limits our ability to qualitatively distinguish between different models of stem cell fate choice based on clone size data only.

**Decision letter after peer review:**

Thank you for submitting your article "Universality of clonal dynamics poses fundamental limits to identify stem cell self-renewal strategies" for consideration by *eLife*. Your article has been reviewed by two peer reviewers, and the evaluation has been overseen by a Reviewing Editor and Naama Barkai as the Senior Editor The reviewers have opted to remain anonymous.

The reviewers have discussed the reviews with one another and the Reviewing Editor has drafted this decision to help you prepare a revised submission.

Summary:

This manuscript provides an important contribution to the field of stem cell biology regarding their 'division modes' (symmetric vs. asymmetric divisions), as detailed in individual comments from reviewers.

Essential revisions:

The reviewers felt that the concerns can be mainly addressed textual changes as detailed in individual comments.

Reviewer #1:

This manuscript describes a general framework of understanding fate decision models, which have been previously proposed in explaining the dynamics of tissue stem cells in homeostasis. The work makes a solid contribution especially in clarifying how a broad class of models conceptually and quantitatively fits with the population asymmetry or the invariant asymmetry model. Such universality argument may have been already taken for granted by physicists/mathematicians but may have not been obvious to biologists who have specific systems in mind that have much richer cell state structure than the original simplified models.

The main criticism I have about the work is that the manuscript may misguide the readers to think that the GIA and GPA models are the only models that have been discussed in explaining the clonal statistics. As one of the authors showed in their previous paper (Greulich and Simons, 2016), the basic statistics such as the clone size distribution will change when adding spatial regulation to the voter model class, which is distinct from what is obtained by GIA or GPA. In fact, for all the tissues that have been studied so far, including the gut (10.1016/j.cell.2010.09.016, Lopez-Garcia et al., 2010), seminiferous tubule (Klein et al., 2010, Hara et al., 214, and 10.1016/j.stem.2018.11.013), and the skin (10.1016/j.stem.2018.09.005), experiments suggest that the spatial correlation between the fates are non-negligible and therefore the asymptotic statistics should follow the voter model. I would assume that there needs to be at least a justification in only studying the cases without spatial interactions here, especially if the authors do not know any specific example that nicely corresponds to their model.

Reviewer #2:

This work builds on several recent high-profile applications of statistical physics to infer the mode of stem cell fate choice in mammalian tissues based on clone size distributions obtained via cell lineage tracing.

The authors note that two conceptually distinct 'models' of stem cell fate choice, the Population Asymmetry (PA) and Dynamic Heterogeneity (DH) models, exhibit the same asymptotic clone size distributions. They then consider a more general set of possible models for the proliferation, differentiation and loss of a multi-type population of cells in a tissue.

Through a detailed mathematical analysis, the authors demonstrate that this large set of models can, in homeostatic tissues, be decomposed into two 'universality classes' in terms of their predicted asymptotic clone size statistics. This limits our ability to qualitatively distinguish between different models of stem cell fate choice based on clone size data only.

Overall I found this work, though technically complex, to be well motivated and well explained. The authors provide a significant advance to our understanding of how different descriptions of stem cell fate choice may be qualitatively distinguished experimentally (though, as the authors note, we may nevertheless apply parameter inference techniques and model selection to quantitatively distinguish competing models based on given datasets).

---

## [Author Response]

Reviewer #1:[…] The main criticism I have about the work is that the manuscript may misguide the readers to think that the GIA and GPA models are the only models that have been discussed in explaining the clonal statistics. As one of the authors showed in their previous paper (Greulich and Simons, 2016), the basic statistics such as the clone size distribution will change when adding spatial regulation to the voter model class, which is distinct from what is obtained by GIA or GPA. In fact, for all the tissues that have been studied so far, including the gut (10.1016/j.cell.2010.09.016, Lopez-Garcia et al., 2010), seminiferous tubule (Klein et al., 2010, Hara et al., 2014, and 10.1016/j.stem.2018.11.013), and the skin (10.1016/j.stem.2018.09.005), experiments suggest that the spatial correlation between the fates are non-negligible and therefore the asymptotic statistics should follow the voter model. I would assume that there needs to be at least a justification in only studying the cases without spatial interactions here, especially if the authors do not know any specific example that nicely corresponds to their model.

We thank the reviewer for this remark. The reviewer notes correctly that in most tissues cell-extrinsic regelation via cell-cell signalling determines fate choices. This can indeed be represented by a (multi-type) voter model – a lattice model where lost cells are compensated by dividing neighbours. However, it has been shown that a voter model in any dimension larger than one has the same asymptotic rescaled clone size distribution as the corresponding branching process without interaction, which is the type of model we use (shown rigorously by Bramson and Griffeath, 1980, and discussed in the context of cell fate dynamics by Klein and Simons, 2011). For mean clone sizes as function of time *t*, there is no difference for dimensions larger than two (in the two dimensional scenario merely a logarithmic prefactor distinguishes the mean clone size in both models; since this factor is nearly constant for large *t*, it often cannot be distinguished by experimental data (Klein and Simons, 2011)).

Therefore, while a voter model may be a more realistic depiction of a tissue with spatial cell-cell interaction, the asymptotic rescaled clone size distribution of such a model is the same as for a corresponding model without spatial regulation, if considering tissues with cell arrangements of dimensions larger than one. Hence, our results would not be different for a corresponding voter model where spatial regulation is included, for two- or three-dimensional tissues such as epithelial sheets and volumnar tissues (e.g. liver or stroma). Yet, we agree that for quasi one-dimensional arrangements of stem cells, as found in seminiferous tubules and intestinal crypts, the clone size distribution differs, and these are indeed not covered by our model.

In the original version of the manuscript we expressed those limitations after the model definition: "This model is a general multi-type branching process, which is suitable to describe cell population dynamics in any dimension larger than one, even under cell-extrinsic regulation (Klein and Simons, 2011; Bramson and Griffeath, 1980)", but we acknowledge that this point warrants a clearer explanation. We therefore now include at this point a paragraph that clarifies the validity range of results to be expected from our model. Furthermore, we now discuss this in the Discussion section.